# A nanoscale robotic cleaner

Jin Qin[1] ✉, Carsten Büchner[1], Xiaofei Wu ®[2] & Bert Hecht ®[1] ✉

Photon-recoil–based actuation enables maneuvering of micro- and nanoscale objects without beam steering or tight focusing, mitigating system complexity and photodamage. Recent light-driven microdrones achieved full control in two dimensions using multiple laser fields; however, for many applications, sacrificing degrees of freedom allows substantial miniaturization and improved propulsion efficiency. Here, we demonstrate sub-micrometer nanorobots actuated by a plasmonic directional antenna that simultaneously provides propulsion force and orientation control. The nanorobots reach propulsion speeds up to 50 µm/s, with their motion direction intrinsically locked perpendicular to the linear polarization axis. Circularly polarized light pulses lift the resulting twofold orientational degeneracy through spin–momentum transfer. Using opto-thermophoretic forces, nanorobots efficiently capture, transport, reversibly assemble, and release bacteria. By sequencing linear and circular polarization states, they execute complex, high-precision trajectories to systematically sweep defined regions, functioning as light-driven robotic cleaners. This work expands the capabilities of nanorobots for biological manipulation and high-speed, localized sensing.

The interaction of light with resonant optical nanostructures can induce transfer of linear or angular momentum, generating optical thrust or torque. At the micro- and nanoscale, the inherently low mass and moment of inertia of such objects result in surprisingly large accelerations. This principle has been widely explored for the actuation of objects, e.g., to induce rotation of nanoparticles and nanowires by spin or orbital angular momentum transfer[1–7], as well as to propel objects through unidirectional light scattering[8–10], yet failing to control both rotational and translational motion. Tanaka et al[11] used a plasmonic dimer that generates thrust via asymmetric scattering. However, efficient translational motion required orientation control to mitigate Brownian rotation. To this end, a line-focused laser trapped and aligned the structure via gradient forces—an approach that limits maneuverability and increases system complexity. To overcome these constraints, we recently proposed plasmonic microdrones that can be actuated independently of their orientation[12,13]. Our design features four chiral resonant plasmonic antennas that can be individually addressed to generate unidirectional scattering under circularly polarized light illumination. By modulating the intensity and polarization properties of two laser beams, the

microdrone achieved full control over all three independent degrees of freedom in 2D motion, completely eliminating the need for confinement via gradient forces.

However, steering a functional micro- or nanorobot within a 2D plane does not necessarily require full control over all three degrees of freedom. A simplified approach, also used in conventional ground vehicles, such as cars, involves generating thrust in one direction while controlling the vehicles orientation and stabilizing it by a self-correcting mechanism. This concept has been theoretically proposed for the self-stabilized photonic levitation of nanostructured objects[14] and experimentally realized for a bulky "metavehicle" covered with a dielectric metasurface[15]. The metavehicle rotates via angular momentum transfer under circularly polarized light, whereas directional diffraction of linearly polarized light generates optical thrust forces. Additionally, the linearly polarized light induces a self-correcting torque that locks the metavehicle's orientation to maximize propulsion efficiency. Self-correcting torques[1] were also used in optical tweezers to control rotation by adjusting the direction of linear polarization[3,4]. However, while effective at larger scales (-12 µm × 10 µm × 1 µm), the metasurface approach does not lend itself to further miniaturization, since the action of a

[1]Nano-Optics and Biophotonics Group, Experimentelle Physik 5, Physikalisches Institut, Universität Würzburg, Am Hubland, Würzburg, Germany. [2]Leibniz Institute of Photonic Technology, Albert-Einstein-Straße 9, Jena, Germany. ✉e-mail: jin.qin@uni-wuerzburg.de; bert.hecht@uni-wuerzburg.de

metasurface requires a periodic arrangement of elementary structures.

Here, we demonstrate light-driven nanorobots that are smaller than 1 μm and exhibit a more than 1000 times lower mass than the metavehicle. They are based on single plasmonic motors that optimize both unidirectional scattering to generate optical thrust and self-correcting torque to suppress Brownian rotation.

We first validate the concept by studying a microrobot equipped with a plasmonic nanomotor configuration that optimizes orientation locking and propulsion separately (see Fig. 1a, left). We then continue miniaturization, creating a nanorobot using a more compact motor design that combines both asymmetric scattering and orientation-locking (see Fig. 1a, right). Figure 1b demonstrates an example of steering a nanorobot along an elementary rectangular trajectory using a sequence of linearly polarized and intermittent clockwise (CW) circularly polarized illumination. The latter is applied for a short time after the nanorobot reaches the corners of the rectangular path. This ensures that the nanorobot reorients sufficiently to be captured in the correct torque potential minimum before continuing its linear motion, which is achieved solely through linear polarization. Based on this principle, we demonstrate deterministic and precise steering of nanorobots along complex trajectories, including the alphabet letters "EP5" and a rectangular spiral pattern. To explore potential biological applications, we immerse both types of robots in aqueous suspensions containing *Escherichia coli* and *Staphylococcus carnosus*, demonstrating their ability to capture, reversibly collect, self-assemble, and dispose of large numbers of bacteria.

## Results
### Working principle of the self-orienting microrobot

The self-orienting micro- and nanorobots include plasmonic motors for propulsion and structures for orientation locking, both consisting of configurations of gold nanorods, as depicted in the SEM images in Fig. 1a. All particles are optimized with respect to their geometries and arrangement and are fabricated by helium-ion-beam milling from monocrystalline gold flakes (50 nm thickness) using the outline-milling technique. After fabrication, the gold nanorod arrangements are embedded into a rigid transparent silica disc with a diameter of 1.5 μm (0.92 μm) and a mass of 0.71 pg (0.26 pg) by means of electron-beam

lithography. The completed micro- and nanorobots are released into an aqueous solution for experiments by dissolving a sacrificial layer.

For the microrobot case (Fig. 1a, left and Fig. 2a), two dimer antennas (top and bottom, red-dashed boxes) are responsible for generating optical thrust forces via directional scattering of photons. They feature nanorods of slightly different lengths (170 nm and 150 nm, respectively) and a uniform width of 60 nm. A strongly asymmetric scattering pattern arises (see lower left inset of Fig. 2b) due to the far-field interference of the two off-resonantly excited dipole scatterers that are spatially separated by a certain distance. This results in constructive interference perpendicular to the rods' long axes on one side and destructive interference on the opposite side[11,16]. Optimization details are provided in the Supplementary Information S4.

The two additional sets of dimer structures (Fig. 2a, blue-dashed boxes, 160 nm length, 60 nm width) are designed to generate self-orienting torques and consist of nanorods of equal length parallel to the nanorods responsible for propulsion. Due to their elongated shape, the induced torque $T_z(\theta)$ is proportional to $\sin(2\theta)$, where $\theta$ is the angle between the nanorod's long axes and the linear polarization direction[17]. We introduce the orientational trapping potential (OTP), defined as:

$$U(\theta_0) = \int_{\theta_0}^{\pi/2} T_z(\theta)d\theta \propto [1 + \cos(2\theta_0)] \tag{1}$$

which characterizes the orientation correction capability and exhibits stable and metastable extreme values. If the nanorod deviates from parallel alignment with the polarization in a stable minimum, a restoring torque is generated. Perpendicular alignment relative to the polarization corresponds to a metastable zero-torque state, which is highly sensitive to external disturbances such as Brownian motion. Once the orientation deviates from this metastable point, the restoring torque drives the nanorod toward the nearest stable minimum as indicated in Fig. 2e. This instability suggests the possibility of using a short pulse of circularly polarized light to nudge the device into the correct orientation. Interestingly, simulations show that the maximum restoring torque does not occur on resonance but rather at a slightly detuned wavelength, where the nanorod's induced polarization is maximized. Optimization details are provided in Supplementary Information S4.

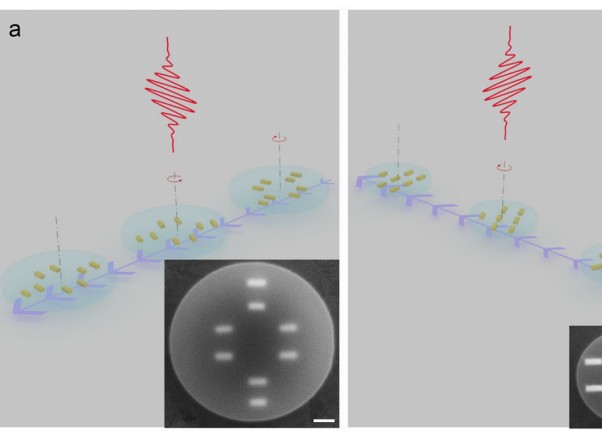
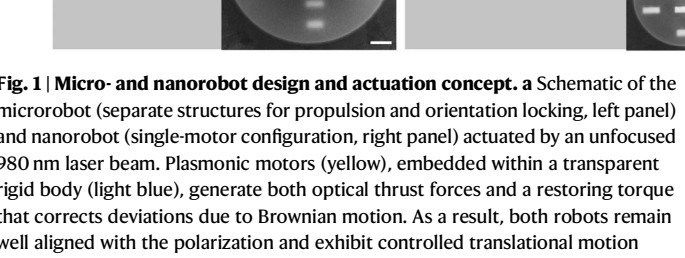
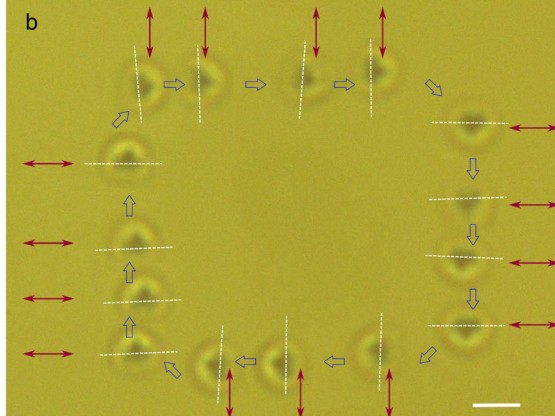

**Fig. 1 | Micro- and nanorobot design and actuation concept. a** Schematic of the microrobot (separate structures for propulsion and orientation locking, left panel) and nanorobot (single-motor configuration, right panel) actuated by an unfocused 980 nm laser beam. Plasmonic motors (yellow), embedded within a transparent rigid body (light blue), generate both optical thrust forces and a restoring torque that corrects deviations due to Brownian motion. As a result, both robots remain well aligned with the polarization and exhibit controlled translational motion (indicated by the straight path with blue arrows). *Inset*: Scanning electron microscopy (SEM) images of the microrobot (left) and nanorobot (right). Scale bar: 200 nm. **b** Superimposed video frames showing the nanorobot following a rectangular trajectory by switching between two orthogonal linear polarizations as indicated by the double arrows (Frame intervals are provided in the Supplementary Fig. S1). The white-dashed lines indicate the orientation of the nanorobot. At each corner, a short pulse of CW circularly polarized light is applied to enforce a right turn. Scale bar: 1 μm.

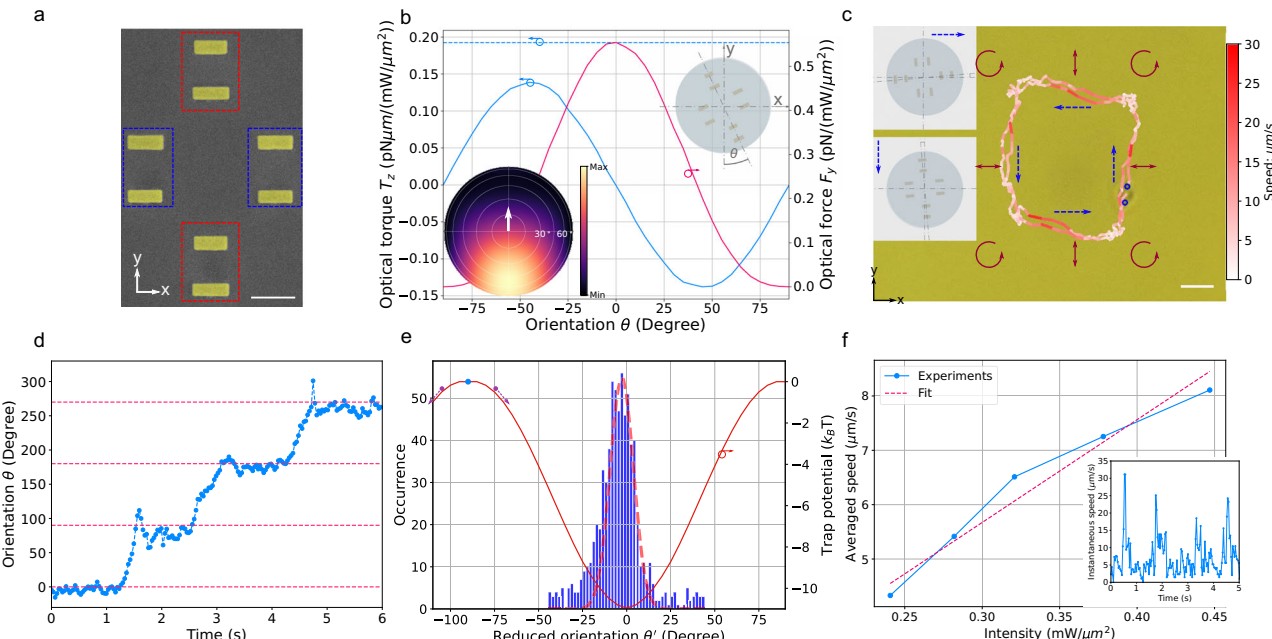

**Fig. 2 | Orientational trapping, thrust force and microrobot steering. a** SEM image of the microrobot showing the plasmonic dimers (red dashed boxes) and self-orienting nanorods (blue dashed boxes). Scale bar: 200 nm. **b** Optical forces $F_y$ and torques $T_z$ as functions of the microrobot's orientation angle $\theta$. The horizontal blue-dashed line indicates the optical torque generated by applying a circularly polarized laser. *Insets*: Definition of the rotation angle $\theta$ (top right, increasing positive values denotes counterclockwise rotation) and representative far-field scattering pattern of a plasmonic dimer (bottom left). **c** Microrobot trajectory under alternating horizontal and vertical linear polarization, with brief circular-polarization pulses applied at turning points (laser intensity: 0.32 mW/µm²). Color indicates instantaneous velocity. Blue-dashed arrows show the motion direction; blue circles mark detected dimer positions. Scale bar: 2 µm. *Inset:* Definition of the reduced orientation angle $\theta'$ used for trajectory analysis. **d** Time-dependent orientation angle $\theta$ extracted from the movie frames in **c** (Third episode in Supplementary Video 1). The pink lines indicate orientation angles of $\theta$ = 0°, 90°, 180° and 270°. **e** Histogram of the reduced orientation angles $\theta'$ in **c** alongside the simulated OTP from **b**. One of the metastable maxima in OTP is marked by the blue dot. When perturbed, the device spontaneously rotates towards the nearest stable minimum, indicated by the purple dots and arrows. **f** Average velocity of the microrobot as a function of the laser intensity recorded while following a rectangular trajectory (Supplementary Video 1). *Inset:* Time-dependent instantaneous translational velocity of the microrobot at a laser intensity of 0.32 mW/µm².

Figure 2b, magenta line, shows that the optical thrust force is maximized when the microrobot's orientation is at $\theta = 0°$, where the major axes of all plasmonic scatterers are aligned with the linear polarization along the x-axis. When the microrobot rotates away from this alignment, the optical force gradually decreases and reaches zero when the major axis is perpendicular to the polarization. The self-correcting torque, due to its $\sin(2\theta)$-characteristic, effectively locks the orientation along the polarization direction, with its maximum value comparable to the torque generated by spin momentum transfer.

A microrobot can be steered efficiently along a rectangular trajectory with high instantaneous velocity (up to 30 µm/s) in an area of 8 µm by 8 µm illuminated by a weakly focused laser beam (power: 100 mW, intensity: 0.32 mW/µm²) at 980 nm. During steering, the microrobot remains stably suspended above the substrate due to the balance between the optical radiation pressure and the repulsive electrostatic force arising from surface charges (Supplementary Information S9.1). Figure 2c outlines the trajectory taken by the microrobot with different hues of red encoding the speed. At the laser intensity used, we estimate an optical thrust force of 0.176 pN and a depth of the orientational trapping potential of up to 10 $k_B T$, as shown in Fig. 2e. The translational motion along the four sides of the rectangular trajectory is obtained by switching between horizontal (HP) or vertical (VP) linear polarization, adjusted by varying the operating voltage of an electro-optical modulator (EOM). Interestingly, during each translational segment, the microrobot's velocity is not uniform; rather, it accelerates until reaching the middle of the path and then decelerates, an effect caused by local laser intensity variations. To make sure that the microrobot is making precise 90°-turns, a pulse

(~100 ms) of counterclockwise (CCW) circularly polarized light is applied each time when the microrobot reaches a corner of the rectangular trajectory. In principle, switching the linear polarization from HP to VP should automatically reorient the microrobot due to the self-correction effect. However, because of the fast switching of the polarization at the corner and the disc's moment of inertia, the microrobot ends up in the metastable maximum of the OTP, and a reorientation can occur in two ways, leading to a 50% chance of a right turn instead of a left turn. Applying a circularly polarized pulse with the appropriate helicity provides additional torque to ensure a deterministic turn. 90°-turns are an important building block for trajectories that systematically scan a certain area and therefore deserve particular attention. Smooth paths with only small rates of change of the propagation angle can be implemented just by turning the linear polarization.

To statistically examine the orientation stabilization effect, we extract the microrobot's orientation from video frames by identifying the positions of the plasmonic motors using an image-processing algorithm akin to what is used in localization microscopy[13] (Detailed process can be found in Supplementary Information S3). After localizing the motors, the orientation angle is determined by analyzing the orientation of the line linking the motors. The change in orientation angle $\theta$ as a function of time while tracing the path in Fig. 2c is plotted in Fig. 2d, where the plateaus correspond to translational motion along a side of the rectangle. Spikes occurring at the corners indicate the action of the above-mentioned re-orientation pulses. The reduced orientation angle $\theta'$ is defined as the line connecting the two plasmonic dimers and the horizontal (or vertical) axis, depending on whether the microrobot is moving horizontally or vertically, as sketched in the top-

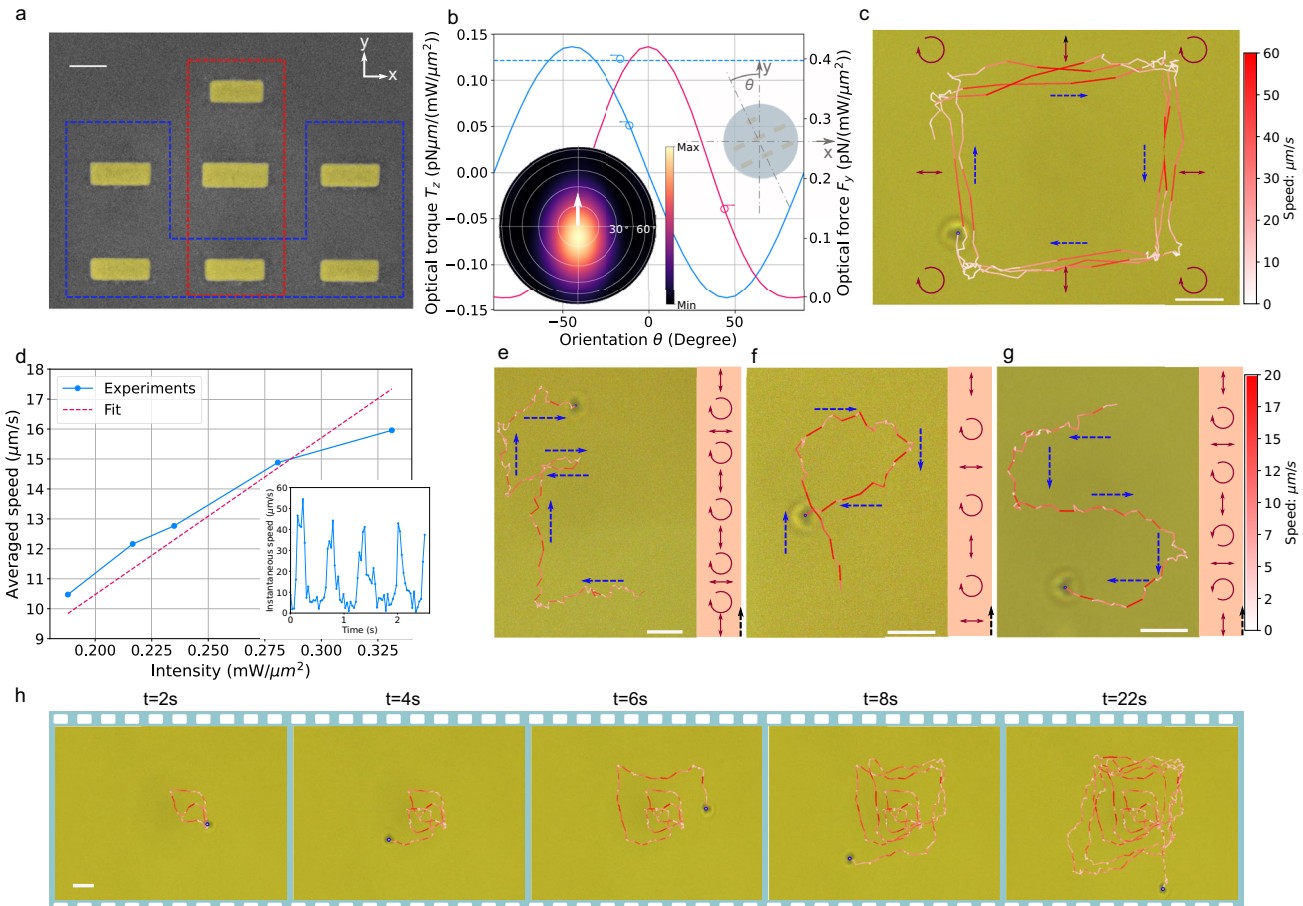

**Fig. 3 | Optical properties and steering of nanorobots. a** SEM image of the nanorobot's plasmonic nanomotor. Propulsion elements and orientation-alignment elements are highlighted by red and blue dashed boxes, respectively. Scale bar: 100 nm. **b** Optical forces $F_y$ and torques $T_z$ as functions of the nanorobot's orientation angle $\theta$. The blue dashed line indicates the optical torque generated by spin momentum transfer. *Insets*: Definition of $\theta$, where positive values indicate counterclockwise rotation (top right), and far-field scattering pattern with the resulting thrust direction (bottom left). **c** Rectangular trajectories of the nanorobot induced by a polarization cycling sequence of horizontal, clockwise circular and vertical polarization indicated by double arrows and circle arrows

(laser intensity: 0.33 mW/µm², last episode in Supplementary Video 2). Color indicates instantaneous velocity; blue dashed arrows show motion direction. Scale bar: 2 µm. **d** Average velocities of the nanorobot as a function of the laser intensities (Supplementary Video 2). *Inset*: Time-dependent instantaneous velocities of the nanorobot in (**c**) for one round trip. **e**, **f** Trajectories forming the characters "E", "P", and "5" (Supplementary Video 3) with corresponding polarization sequences shown to the right (from bottom to top). Scale bar: 2 µm. **h** Spiral rectangular pattern generated by progressively lengthening each linearly polarized state in the cycling sequence (Supplementary Video 4). Panels (**e**–**h**) share a common velocity color scale. Scale bar: 2 µm.

right inset of Fig. 2c. Its distribution for straight path segments only, shown as a histogram in Fig. 2e, follows a Gaussian distribution with a standard deviation of 6.6°. Details are provided in Supplementary Information S5. To complete the analysis of the trajectories, we analyze the velocity of the microrobot as a function of the local laser intensity. By evaluating the center-of-mass positions in each movie frame, we find that the microrobot's average velocity increases linearly with laser intensity, as shown in Fig. 2f, consistent with the characteristics of Couette flow around an obstacle. The instantaneous translational velocity varies significantly, reaching peak values 3 times higher than the average velocity due to the Gaussian-shaped laser intensity distribution. The overall velocity as a function of time is depicted in the inset of Fig. 2f.

**Further device miniaturization and complex motion patterns**
Inspired by the microrobot with separate elements responsible for propulsion and orientation locking, we explore the limits of miniaturization for a functional light-driven and steered microrobot, ultimately leading to a controllable nanorobot. We propose a miniaturized plasmonic nanomotor design as depicted in Fig. 3a, which clearly breaks the 1 µm barrier. We retain the plasmonic dimer configuration

to generate optical thrust via asymmetric scattering (red dashed box) and introduce additional plasmonic nanorods (blue dashed box) to enhance the orientational trapping effect. The optimization process is guided by two key factors: maximizing the optical thrust and the depth of the OTP for $\theta = 0°$.

As a result, we arrive at a nanorobot with a body diameter of 920 nm and a mass of 0.26 pg. Similar to the microrobot, the simulated orientation-dependent optical forces and torques are displayed in Fig. 3b. Notably, the self-correction torque remains comparable to that of the microrobot, indicating that the orientational trapping capability is preserved. Due to its smaller size (≈50% reduced projected area), the nanorobot can achieve even faster peak velocities (up to 50 µm/s) while traversing larger rectangular trajectories within the unfocused laser spot (power: 104 mW, intensity: 0.33 mW/µm²), as shown in Fig. 3c. The expected optical thrust force in this case is 0.132 pN, only marginally smaller than that of its bigger brother. The nanorobot can be maneuvered along rectangular trajectories of 10 µm side length, fully utilizing the 20 µm-diameter laser spot. This is because, as the nanorobot moves through the aqueous solution, the optical thrust force is balanced by the viscous drag force. However, due to the Gaussian profile of the laser spot, the optical thrust force varies with

position. This effect becomes most pronounced when the nanorobot moves near the corners of the rectangular trajectory, where the propulsion velocity decreases further. In Supplementary Information S9.2, we systematically quantify the velocity variation across different positions. An advantage of the high translational velocity is the reduced time required for each motion step, which reduces the influence of Brownian motion. The averaged nanorobot velocity scales linearly with laser intensity, as shown in Fig. 3d. The instantaneous velocity exhibits a square-wave-like pattern, where each peak represents translational motion along the side of the rectangle (inset in Fig. 3d). The velocity profile follows the local intensity of the Gaussian laser spot. The peak instantaneous velocity is nearly five times higher than that of other vehicles propelled by optical thrust forces[11–13,15], which typically exhibit velocities below 10 μm/s. Using computational fluid dynamics simulations, we provide a more detailed analysis of the motion dynamics (for example, the suspended height of the nanorobot) in Supplementary Information S9.2.

Leveraging these capabilities, we demonstrate tracing complex trajectories by dynamically controlling the laser polarization. A series of preprogrammed polarization states and intermittent dwell times with spin-momentum transfer are generated via the corresponding time-sequenced voltage waveforms applied to EOMs, allowing precise control of polarization as a function of time. With this method, we successfully steer the nanorobot along complex trajectories, such as the alphabet letters "EP5" as illustrated in Fig. 3e–g, Supplementary Video 3. Additionally, our nanorobot can function as a "nanoscanner", capable of following a spiral rectangular path to scan a 2D plane, as shown in the sequenced frames in Fig. 3h, Supplementary Video 4.

Deviations from the intended straight path are observed and are attributed to two factors. First, Brownian motion remains the dominant influence. While higher velocities reduce the interaction time and the OTP immediately corrects deviations, the nanorobot's extremely small size and mass also make it more susceptible to random fluctuations induced by Brownian motion. Second, when the device turns by changing the laser polarization, it requires time to adjust its orientation to the correct angle while overcoming a drag torque. During this reorientation, optical forces continue to act on the device, causing it to move along a curved trajectory until its orientation is aligned with the polarization.

## Micro- and Nanorobot interactions with ensembles of bacteria

A compelling demonstration of the possible benefits of the well-controlled maneuverability of our nanorobots in aqueous solutions is their interaction with biological objects. Bacteria are ideal candidates as they are comparable in size to our robots. Due to the presence of multiple plasmonic nanostructures that produce localized optical near fields and mild local heating, our robots can effectively leverage both optical trapping and photo-thermal effects. The optical-trapping effect is confined to the immediate vicinity of the plasmonic structures due to short-ranged gradient forces caused by the localized near-field intensity enhancement, allowing bacteria to be trapped above gold nanorods at the top surface of the robot[13,18]. In contrast, the moderate local heating originating from plasmonic absorption causes the thermophoretic force $F_t$, which is primarily governed by the temperature gradient $\nabla T$[19] and provides a significantly extended trapping range.

As illustrated in the sequential frames in Fig. 4a and Supplementary Video 5, we immerse the microrobot in an aqueous solution containing a mixture of rod-shaped *Escherichia coli* and spherical *Staphylococcus carnosus* bacteria. In the first frame, the microrobot captures a single *Staphylococcus carnosus* on its top surface, an effect attributed to the combined action of optical trapping and thermophoretic force. Interestingly, when the microrobot accelerates in regions of higher local intensity, the captured bacterium temporarily detaches due to liquid drag. This behavior confirms that the bacteria are not adhering to the microrobot's surface but are instead

held by external forces. The detachment due to the liquid drag force also provides a possible means to detach a bacterium without completely switching off the driving laser. In the latter case, detachment occurs, but the distance between microrobot and bacterium only increases slowly, and switching on the laser may again lead to immediate trapping. We demonstrate the microrobot's ability to transport the bacterium along a rectangular trajectory using the methods described above, as shown in Supplementary Video 5.

Over time, the microrobot can accumulate a significant number of bacteria around its body, as seen in the second to fourth frames. These bacteria assemble in a surprisingly regular pattern compatible with assembly of a close-packed lattice of hard spheres. Notably, the assembly process is independent of bacterial shape or type, effectively capturing both *Escherichia coli* and *Staphylococcus carnosus*. The bacterial cluster continues to grow until its diameter reaches the upper limit of the microrobot's capacity, given by the limited range of the attractive thermophoretic force, which eventually leads to an equilibrium of attaching and detaching bacteria due to thermal activation. Despite carrying a bacterial load hundreds of times heavier than itself, the microrobot and its load remain maneuverable, and the microrobot maintains its ability of orientational trapping along the direction of linear polarization. Once the bacterial cluster reaches its maximal size, we turn off the laser. The absence of optical and thermophoretic forces causes the assembled cluster to slowly disperse immediately, with bacteria diffusing away from the microrobot as shown in the last frame. This behavior clearly demonstrates that the assembly is a fully reversible process.

The assembly of bacteria into a cluster of a particular size can be used to quantify the responsible thermophoretic force. In a first step, we excluded the possibility that optical trapping forces play a role in the assembly. We therefore carefully checked that no resonant optical fields, akin to whispering gallery modes[20], can form upon attachment of bacteria to the robot's body and that solely the thermophoretic force is responsible for the bacteria trapping and accumulation. In Supplementary Information S7.1, we present detailed simulations comparing the optical gradient force and the thermophoretic force acting on the assembled bacteria. To simplify the model, we analyze in-plane accumulation only, as illustrated in Fig. 4b. The thermophoretic force can be expressed as: $F_t = -k_B T S_T \nabla T$, where $S_T$ is the Soret coefficient, $k_B$ is the Boltzmann constant, and $\nabla T$ is the gradient of the temperature field[19,21]. The direction of the thermophoretic force is primarily determined by the sign of $S_T$, causing particles to migrate toward warmer regions when $S_T$ is negative and away from warmer regions when $S_T$ is positive. Previous studies have shown that certain biological cells exhibit negative $S_T$, as do DNA[22], bacteria[23], and erythrocytes[21]. However, the exact value of $S_T$ is largely unknown.

In our micro- and nanorobots, plasmonic nanostructures act as localized heat sources, creating a mild temperature increase and a concomitant gradient that is harmless to bacteria. While often seen as a drawback, plasmonic heating is essential here, as it enables strong thermophoretic trapping under moderate laser intensity. Specifically, as shown in Fig. 4c, we estimate the maximum temperature increase $\Delta T$ under varying laser intensities to be less than 10 K even at a laser intensity of 0.4 mW/μm². This ensures compatibility with biological cells while minimizing the risk of photo-thermal damage. Additionally, since the nanoscale gold antennas serve as the sole heating source in our system, they induce a significant temperature gradient $\nabla T$ of up to 0.02 K/nm, causing a comparatively strong thermophoretic force. Particle accumulation due to convective flow[24–26] can be neglected as we confirm by finite element simulations. The predicted flow velocity is extremely small (below 2 nm/s), consistent with previous experimental findings[25,27].

The observed assembly effects induced by thermophoresis can be used to estimate the Soret coefficient $S_T$ of bacteria. As shown in Fig. 4a, we observe that for a fixed laser intensity, the bacterial cluster

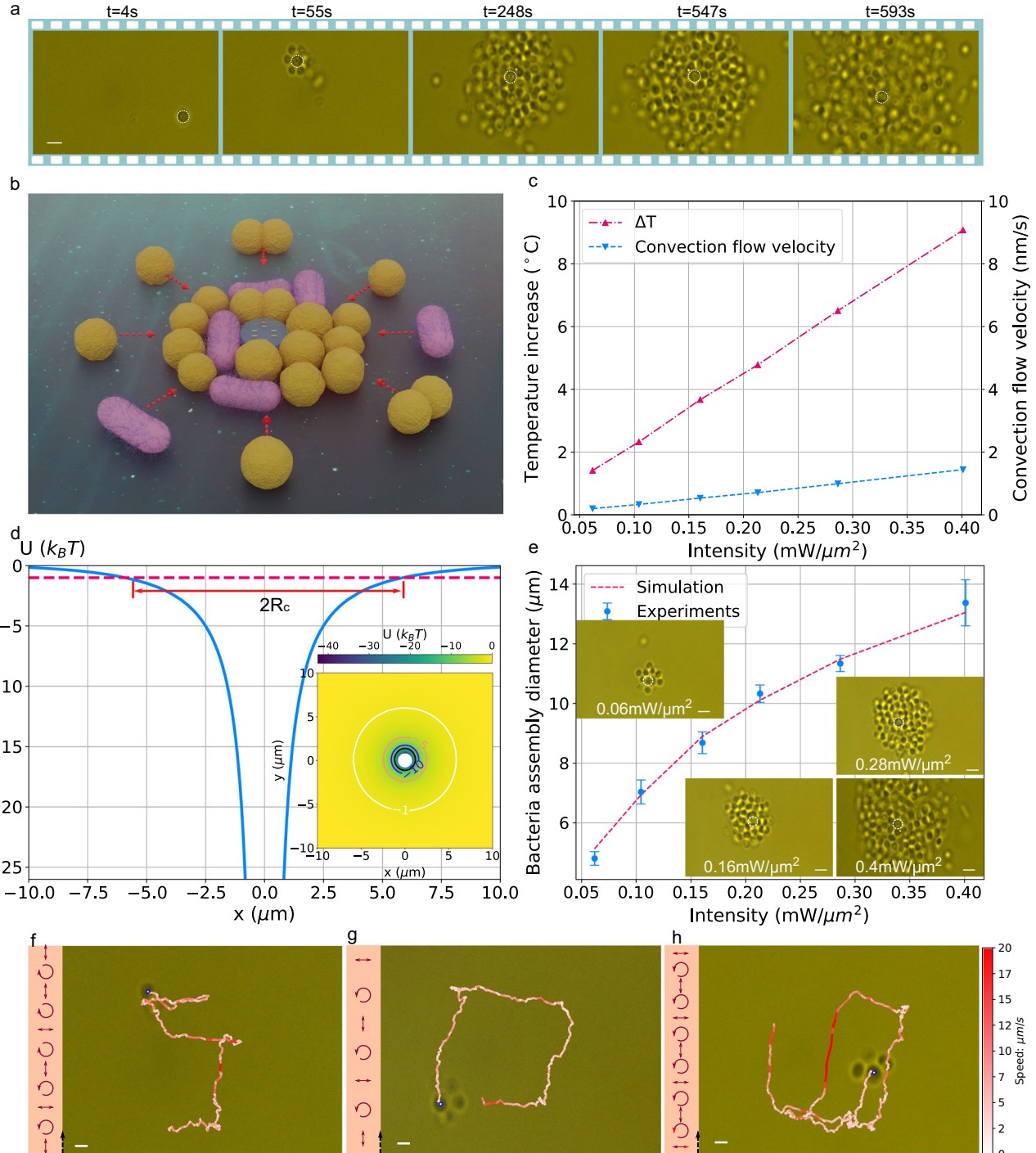

**Fig. 4 | Micro- and nanorobots trap, assemble, transport, and release multiple bacteria and form densely-packed bacterial clusters. a** Video-frame sequences taken from Supplementary Video 5, showing the complete process of trapping and assembling multiple bacteria in a mixed suspension of *Escherichia coli* and *Staphylococcus carnosus* using a single microrobot. The white-dashed circle indicates the microrobot's position. Scale bar: 2 μm. **b** Artistic illustration of bacteria being assembled around the microrobot due to the long-range thermophoretic force (red-dashed arrows). **c** Simulated temperature increases and convection flow velocities for different laser intensities, obtained from multiphysics heat-flow simulations (extraction positions indicated in Fig. S10, Supplementary

Information S10). **d** Effective trapping potential $U$ induced by the thermophoretic force at a laser intensity of 0.286 mW/μm². Position corresponding to a potential depth of $-1\,k_BT$ is marked as the red dashed line, from which $2R_c$ can be extracted. *Inset*: 2D potential map with contours at $-1$, $-5$, $-10$, and $-15\,k_BT$. **e** Equilibrium bacterial cluster diameter as a function of applied laser intensity. *Inset*: Bacterial aggregation images at different laser intensities. **f** Nanorobot transporting a single *Escherichia coli* bacterium along a "5"-shaped trajectory. Scale bar: 2 μm. **g, h** Nanorobot assembling multiple *Staphylococcus carnosus* bacteria and carrying them along a rectangular trajectory g and a "6"-shaped trajectory h (Supplementary Video 6). Scale bar: 1 μm.

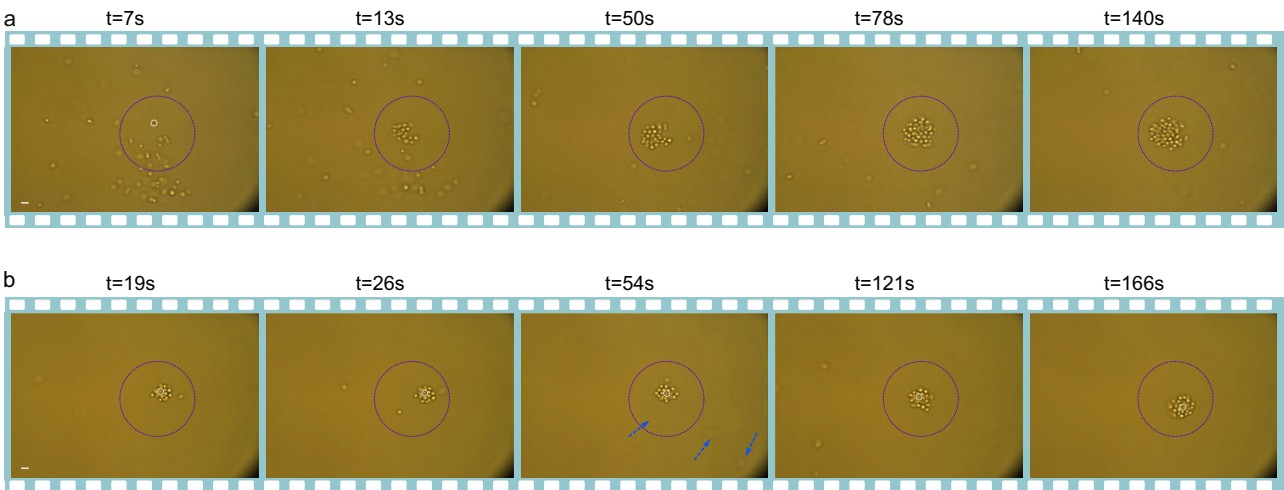

**Fig. 5 | Micro- and nanoscale robotic cleaner. a** Nanorobot (white dashed circle in the first frame) is maneuvered to collect bacteria from the solution (Supplementary Video 7). By adjusting the path via laser polarization and controlling the stage, bacteria are effectively assembled around the nanorobot, resulting in a bacteria-free target area in the final frame. **b** A microrobot (white dashed circle) is maneuvered to collect bacteria from a solution with a lower bacterial concentration (Supplementary Video 8). The microrobot not only gathers bacteria from its immediate surroundings but also attracts bacteria suspended at different heights, as indicated by faint spots marked with blue dashed arrows in the third frame. In the final frame, nearly all bacteria have been successfully collected. Scale bar: 2 μm. The purple dashed circle indicates the laser spot size.

ceases to grow after reaching a certain diameter. This happens when, at the outermost rim of the cluster, the effective trapping potential due to the thermophoretic force is equal to 1 $k_B T$, equal to the average kinetic energy of bacteria undergoing Brownian motion. We therefore varied the laser intensity and measured the resulting bacterial cluster diameters, $R_c$, as a function of the laser intensity. The results are displayed in Fig. 4e. The trapping potential generated by the thermophoretic force is simulated based on temperature distributions obtained from finite-element simulations of heat transfer. By identifying the trapping region's radius $R_c$ corresponding to an energy threshold of 1 $k_B T$, we fit the simulation results to experimental data, yielding an estimated $S_T$ value of $-2.45 K^{-1}$. Although the effective value can depend on factors such as bacterial size, anisotropy, surface characteristics, and fluid conditions, it nevertheless offers a reasonable approximation derived from the trapping-equilibrium region. Further analysis and simulations of the effective Soret coefficient are presented in Supplementary Information S8.4.

Notably, this reversible bacterial assembly effect is not exclusively observed for the microrobot. Experiments conducted with the nanorobot exhibited the same behavior. However, for clearer visualization, we primarily present results obtained using the microrobot, as its four-motor configuration allows for better imaging in the presence of bacterial clusters. Additionally, the thermophoretic trapping effect is not confined to the xy-plane; a temperature gradient in the xz-plane can also trap nearby bacteria. As shown in Supplementary Video 5, some bacteria migrate toward the microrobot and become trapped above the device. A more detailed analysis of this three-dimensional trapping behavior is provided in Supplementary Information S7.2.

Importantly, even when carrying one or multiple bacteria, the maneuverability of our robots remains well preserved. In Fig. 4f, we demonstrate a nanorobot transporting a single *Escherichia coli* bacterium along a trajectory shaped like the number "5". In Fig. 4f–g, the nanorobot, now carrying multiple bacteria, successfully follows both a rectangular and a "6"-shaped trajectory. Interestingly, even with a bacterial load, the nanorobot maintains a certain velocity, demonstrating the robustness of the driving mechanism in inhomogeneous environments and effective movement with a sufficiently low drag coefficient under different loading conditions. However, when the robots carry bacteria, their translational velocity $v$ becomes limited by the increased hydrodynamic drag acting on both the robot and the attached bacterial load. In this case, the velocity satisfies $v \le F_o / (N \times 6\pi\eta r + D)$, where $F_o$ is the optical thrust generated by the plasmonic motors, $\eta$ is the dynamic viscosity of water, $r$ is the effective hydrodynamic radius of the robot, $N$ is the number of trapped bacteria, and $D$ is the additional drag coefficient associated with the robot alone, which is extracted from computational fluid dynamics (CFD) simulations. In Supplementary Information S7.3 and S8.1, we provide a detailed analysis of the velocity constraints for both robotic platforms when carrying bacterial cargo, showing that the achievable propulsion speed has an upper bound determined by the total drag. Importantly, the orientation-alignment mechanism remains effective even under heavy loading, as discussed in Supplementary Information S8.2. Because the trapped bacteria do not fully adhere to the robot surface, likely due to electrostatic interactions, they do not significantly disrupt the optical orientational torque. This behavior is confirmed in Supplementary Video 5, even when the robot carries a large bacterial cluster and its translational velocity is substantially reduced by viscous drag, its orientation still reliably follows the laser polarization.

## Discussion

Inspired by the above demonstrations, we can conceptualize our micro- and nanorobots as robotic cleaners for biological applications. As shown in Supplementary Video 7 and Fig. 5a, our nanorobot is capable of collecting a cluster of bacteria, transporting them and disposing of them at a designated location. Due to the limited size of the laser spot, we extend the operational range by integrating stage movement. Specifically, when the nanorobot approaches the edge of the laser spot, we shift the stage to reposition the nanorobot back to its starting point. This method significantly expands the effective working area, allowing us to efficiently gather and relocate bacteria to a single designated region. With this method, the robot can dynamically sweep a much larger area of the solution compared with a static cleaner, resulting in a substantial improvement in cleaning efficiency. In Supplementary Information S8.3, we provide a more detailed comparison between static robots and mobile robots.

Another demonstration, also shown in Supplementary Video 8 and Fig. 5b, involves using a microrobot to collect bacteria at varying heights within the suspension. When the microrobot, already carrying

a bacterial cluster, approaches free-floating bacteria, the thermophoretic force attracts and traps additional bacteria from different heights around the robot. Once loaded, we can then steer the microrobot to a different location, effectively clearing the targeted area of bacteria.

Compared with conventional optical tweezers, which can also be used to manipulate bacteria such as *Escherichia coli*[28], the laser intensity used in our experiments is two orders of magnitude lower than that required for optical trapping of bacteria. Additionally, our approach is nearly harmless to biological cells, as the overall temperature increase remains below 10 K and is localized to a very small volume. Moreover, unlike optical tweezers, which typically capture single bacteria with one focus point, our robots efficiently assemble multiple bacteria around them, significantly enhancing throughput and processing capability.

Interestingly, for bacteria such as *Staphylococcus carnosus*, which typically adhere to each other when forming clusters, our experiments demonstrate that they can be reversibly assembled and disassembled. This effect is likely due to Coulomb repulsion caused by surface charging of bacterial membranes in solution.

Currently, the translational motion of our robots is restricted to two orthogonal directions (horizontal and vertical) due to the binary switching between HP and VP in our setup. In principle, a turnable linear polarizer could be implemented to enable arbitrary translational motion in the 2D plane by leveraging the orientation-locking effect, thereby enhancing the robots' maneuverability. However, the rotation speed of the linear polarizer cannot be increased arbitrarily, as the robot's orientation cannot follow rapid polarization changes instantaneously (as described in our theoretical model in Supplementary Information S6). Exceeding this limit leads to orientation slip, resulting in nondeterministic turning behavior. In contrast, short pulses of circularly polarized light are better suited for rapid reorientation. This consideration underlies our demonstration design: because the robot operates at relatively high velocities, prompt and reliable orientation changes are essential. Moreover, achieving horizontal and vertical translation is a fundamental requirement for a "scanner" and is sufficient to cover the entire area within a 2D plane. Combinations of horizontal and vertical translations can also approximate any curved path.

In the future, the collective operation of multiple robots will be feasible by integrating spatial light modulators (SLMs) or digital micromirror devices (DMDs), which can divide a single illumination source into multiple independently controlled sub-fields. Each sub-field can be assigned a distinct polarization state and intensity profile, enabling simultaneous and parallel control of several robots. This approach, however, also introduces significant hardware challenges, including the need for highly uniform and stable optical fields, sufficient laser power, and a high-bandwidth control system.

In our previous work[12,13], we introduced a drone-like microrobot designed to achieve controlled propulsion in fluidic environments where Brownian fluctuations perturb both translational motion and rotational orientation. Both that earlier system and the present vehicle-like nanorobot address the same fundamental challenge: reliable navigation at the microscale requires not only unidirectional thrust generation but also stable orientation control.

In the drone-like architecture, two circularly polarized beams with opposite helicities produced counteracting optical torques, enabling active stabilization of the robot's orientation during propulsion. In contrast, the vehicle-like nanorobot employs a single plasmonic directional antenna driven by a linearly polarized beam. This configuration provides both unidirectional photon-recoil thrust and a passive orientational trapping mechanism that aligns the robot with the polarization axis, thereby eliminating the need for multi-beam torque balancing. More detailed comparisons and design trade-offs between the two systems are provided in Supplementary Information S2.

In summary, we have demonstrated a nanoscale robotic system driven by an unfocused laser beam. By combining unidirectional optical thrust due to asymmetric light scattering with orientation-locking effects in linear polarization, our plasmonic motor-based design enables miniaturization of the robot's body to a diameter of approximately 920 nm. This nanorobot can be steered in a 2D plane, reaching high velocities, significantly reducing the impact of Brownian motion. Through precisely designed polarization sequences of the driving light field, our nanorobot can trace complex trajectories with high-speed motion bursts (up to 50 μm/s), such as the path forming the alphabetic letters "EP5". Another key feature of our system is its ability to assemble, transport, and release multiple bacteria simultaneously through a combination of optical trapping and thermophoretic forces, where the latter dominates at larger distances and funnels small objects towards the nanorobot. Remarkably, our robots maintain excellent maneuverability even while carrying bacterial loads of hundreds of times their own weight. This capability allows our nanorobotic system to function as a nanoscale biological cleaner, efficiently collecting bacteria in a specific region and relocating them without causing photodamage. The versatility and precision of our design support a broad range of potential applications in bioengineering, targeted drug delivery, and nanoscale local sensing.

## Methods

### Fabrication

Fabrication starts by synthesizing mono-crystalline gold platelets on glass coverslips[29,30]. A gold platelet with a thickness of 50 nm is selected by fitting the transmittance spectrum resulting from white-light illumination. The glass substrate, coated with 700 nm of indium tin oxide (ITO), is thoroughly cleaned using an ultrasonic bath with acetone and isopropanol. A thin layer (100 nm) of hydrogen silsesquioxane (HSQ) is then spin-coated onto the ITO surface for planarization, followed by the transfer of the gold platelet on top.

The plasmonic motors are fabricated using helium-focused-ion-beam milling (Zeiss Orion NanoFab), where only the outlines of the structures are milled away to isolate them from the gold platelet. Subsequently, after peeling off the gold platelet and leaving behind the outlined nanomotor structures, another 150 nm layer of HSQ is spin-coated on top of the structures. The microrobot body is then defined using electron beam lithography (EBL) and developed by wet chemical etching. To release the microrobots from the glass substrate, the underlying ITO layer is etched away using a hydrochloric acid (HCl) solution.

After a washing step, the cleaned robots are sealed in a liquid cell consisting of a thin polydimethylsiloxane (PDMS) spacer (165 μm thick) with a central hole (6 mm in diameter). For bacterial assembly experiments, an aqueous bacterial suspension is used to fill the liquid cell. The robots are finally detached from the substrate using mild ultrasonication for 5 s.

### Numerical simulations

Finite-element multiphysics simulations are performed using commercial software (COMSOL Multiphysics). The electromagnetic (EM) module is first employed to optimize the optical force and torque of plasmonic gold nanorod assemblies under plane wave excitation. The dielectric function of mono-crystalline gold is obtained from Olmon et al[31]. The refractive indices of water and the HSQ body are set to 1.33 and 1.45, respectively. Scattering boundary conditions are applied to all outer boundaries to minimize EM reflections. Optical forces $\mathbf{F}$ and torque $\mathbf{T}$ are calculated by integrating the time-averaged Maxwell's Stress Tensor $\left\langle \overleftrightarrow{M} \right\rangle$ over the robot's body, using the equations: $\mathbf{F} = \oint_{\mathbf{S}} \left\langle \overleftrightarrow{M} \right\rangle \cdot \hat{n} \, d\mathbf{S}$ and $\mathbf{T} = \oint_{\mathbf{S}} \mathbf{r} \times (\left\langle \overleftrightarrow{M} \right\rangle \cdot \hat{n}) d\mathbf{S}$. Then the EM simulations are placed into a significantly larger domain (100 μm × 100 μm × 100

μm) to simulate heat transfer and laminar flow (HT-LF), mimicking the robot's real working environment in a liquid cell. The HT-LF boundaries are assigned a prescribed temperature of 20 °C and a no-slip wall condition. Detailed simulation parameters for HT-LF can be found in Table S3.

### Optical setup and measurements

A schematic of the optical setup is shown in Fig. S2. The laser used for steering the robots is a continuous-wave laser diode (Thorlabs, BL976-PAG900) with a maximum output power of 900 mW. The laser power and polarization are controlled by two electro-optic modulators (EOMs, Thorlabs, EO-AM-NR-C2), which introduce phase shifts along two orthogonal optical axes. Driving signals are generated by a multifunctional I/O device (National Instruments, USB-6343) and sent to high-voltage amplifiers (Thorlabs, HVA200) with a fixed gain of -20. By adjusting the applied voltages, the laser power can be tuned from zero to its maximum, while the polarization state can be switched from horizontal (HP) to vertical (VP).

The incident laser is loosely focused onto the sample using a polarization-conserving objective (Zeiss, EC Epiplan-Neofluar 20 × /0.5 Pol). The sample is mounted on a piezo stage capable of nanoscale motion control. Images of the sample are collected by a high-NA oil immersion objective (Nikon, MRD01991 CFI Apochromat 100× Oil, NA: 1.49). A CMOS camera (DFK 37AUX252) records images at 30 fps.

### Bacteria culturing

For the experiment, the bacteria *Escherichia coli* (Nissle 1917, MutaFlor) and *Staphylococcus carnosus*[32] (TM300, Microbiology Chair) are cultivated overnight at 37 °C and 180 rpm in tryptic soy broth media (TSB). The bacteria are washed three times with sterilized water. Then, the two types of bacteria are mixed and diluted with Milli-Q water at a ratio of 1:5000.

### Data availability

All data needed to evaluate the conclusions in the paper are present in the paper and/or the Supplementary Materials. The raw data is available at: https://doi.org/10.5281/zenodo.18475358

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

### Acknowledgments

We thank Linus Wilm (NanoStruct GmbH, Würzburg, Germany) for preparing the bacteria. *Staphylococcus carnosus* TM300 was a gift of Friedrich Götz (Tübingen) and was kindly provided by Martin Fraunholz (Chair of Microbiology, JMU, Würzburg). This project was funded by the German Research Foundation (DFG grant no. HE5618/10–1) and the "Staatsministerium für Wissenschaft und Kunst" of the state of Bavaria

within the framework "Hightech Agenda Bayern Plus", specifically, the project "Integrated Spin Systems for Quantum Sensors (IQ-Sense)" which is part of the "Munich Quantum Valley".

## Author contributions

J.Q. and B.H. conceived the project. J.Q. performed numerical simulations and structure optimizations. J.Q., C.B., and X.W. fabricated the samples and constructed the optical setup. J.Q. carried out the optical measurements and analyzed the data. J.Q. wrote the manuscript. All authors contributed to the discussion and manuscript preparation.

## Funding

## Competing interests

The authors declare no competing interests.
