## [Transparent Peer Review file · Nature Communications]

A nanoscale robotic cleaner

Corresponding Author: Professor Bert Hecht

Version 0:

Reviewer comments:

Reviewer #1

(Remarks to the Author)

The manuscript presents a light-driven micro/nanorobot, miniaturized to sub-micrometer dimensions, that functions as a robotic cleaner for biological applications. The robot is powered and steered by a single-wavelength unfocused laser beam using a simplified control architecture where two sets of plasmonic antennas provide both propulsion and orientation control. The robot's direction of motion is locked perpendicular to the driving linear polarization, while deterministic turns are executed using short pulses of circularly polarized light. Leveraging a combination of optical trapping and long-range thermophoretic forces, the robot can efficiently capture and assemble large clusters of bacteria, which can then be released at designated locations. The work is supported by FEA simulations, along with experimental validation demonstrating high-speed maneuvering up to 50 $\mu\text{m/s}$.

The concept of a sub-micron, light-driven robotic cleaner is compelling, and the simplified, miniaturized design is a smart approach. However, the manuscript's impact could be substantially improved. The primary weakness is the lack of a direct comparison to the authors' own previously reported chiral antenna microdrones (refs 1, 13) to clarify the pros and cons of this new design. Additionally, the manuscript needs a more rigorous analysis of the trapping mechanism and a quantitative assessment of the robot's maneuverability and cleaning efficiency. The work requires major revisions to address these critical questions.

Major Comments:

1. The manuscript does not provide a clear comparison to the authors' highly relevant prior work on chiral antenna microdrones. While the new design is clearly smaller, a detailed discussion of the trade-offs is missing. The authors should provide an explicit comparison of the pros and cons of this new "car-like" design versus the previous "drone-like" one, clarifying if the smaller size is a direct consequence of the simplified control scheme and how the propulsion efficiency compares quantitatively to the multi-motor chiral designs.
2. The manuscript also focuses exclusively on 2D motion and trapping but fails to address the robot's stability in the third dimension. While the self-correcting torque prevents in-plane rotation, there is no discussion of what prevents the disc-shaped robot from tilting or tumbling out of the xy-plane, especially during high-speed motion and turns. The authors must clarify the mechanism responsible for maintaining out-of-plane stability, such as the role of vertical radiation pressure or hydrodynamic forces.
3. The analysis of the trapping mechanism is insufficient for a dynamic "cleaner." The paper establishes an equilibrium cluster based on a static ~ 1 kBT potential, but this does not account for the hydrodynamic drag force that trapped bacteria will experience during motion. A deeper potential well is required to retain the cluster, and the manuscript needs a thorough discussion of assembly performance and stability under dynamic conditions. The trapping is also a 3D process, as the robot collects bacteria from different heights, yet the potential analysis in Figure 4d is limited to the 2D xy-plane. The authors must provide a cross-sectional (e.g., xz-plane) analysis of the thermophoretic trapping potential to properly characterize the 3D trapping volume and strength.
4. The claims about the robot's performance are not sufficiently supported by quantitative data. The assertion that the robot "remains fully maneuverable" after collecting a large bacterial cluster is a significant overstatement. The increased hydrodynamic drag must impact performance, and the authors need to provide a quantitative analysis comparing key metrics

like maximum speed and turning precision before and after bacterial assembly. It is also unclear how active motion improves cleaning efficiency over a static trap with a long capture range. A clear rationale, ideally supported by data comparing the clearance rates of a mobile versus a static robot, is needed to justify the "cleaner" concept.

5. All demonstrations show a single robot. For practical applications like large-area cleaning, the authors should discuss the feasibility of collective operation. Potential limitation should be acknowledged.

Minor Comments

1. The overall readability is hindered by the manuscript's structure. The captions for Figures 2, 3, and 4 are excessively long and contain methodological details and interpretations that belong in the main text, disrupting the narrative flow. It is strongly recommended that the authors move these details to the main body of the text and revise the captions to be concise descriptions of the data.

2. Several figures also lack the necessary clarity for straightforward interpretation. The rendered schematic in Figure 1a is difficult to read and should have a more transparent body to show the internal antennas. The caption for Figure 1b should state the time interval between the superimposed frames to provide a sense of speed. Due to the motor's high symmetry, a coordinate system or an arrow defining the "+y" direction of propulsion must be added to Figures 2a and 2b to resolve ambiguity.

3. Some important technical details are also missing from the main text. For instance, the dimensions of the dimer for the orienting torque cannot be found in the main manuscript and should be included for completeness.

Reviewer #2

(Remarks to the Author)

The paper proposes a mechanism that uses a restoring torque induced by linear polarized light to achieve orientational locking, which stably aligns the robot along the linear polarization axis to maximize propulsion efficiency. I think some issues need to be answered or solved before considering the publication of this submission.

The authors state that a brief circularly polarized light pulse is required to achieve precise 90° turns. The need for this auxiliary pulse seems to contradict, to some extent, the core advantage of the "self-alignment" mechanism, which is to control direction using only linear polarized light.

From a deeper physical perspective, this phenomenon reflects the dynamic relationship between the optical torque and the robot's intrinsic angular momentum and inertia. When the polarization state is switched rapidly (approx. 1 ms), the robot's moment of inertia may prevent it from immediately capturing the new stable alignment direction, instead causing it to settle in a metastable maximum of the orientational potential well. In this state, any minor disturbance, such as Brownian motion, could cause the robot to rotate randomly to the left or right, leading to a loss of deterministic control over the turning direction. The authors use a circularly polarized light pulse to provide an additional, deterministic torque via spin-momentum transfer, overcoming this uncertainty and ensuring the robot enters the correct stable alignment state each time.

Although this turning mechanism is effective, its potential limitation lies in controlling more complex and smooth trajectories. The paper mentions in the discussion that "smooth paths can be implemented just by turning the linear polarization," which implies two different control modes: one for straight motion and slow turns using pure linear polarization control, and another for precise 90° turns using a hybrid of linear and circular polarization. To make the paper more rigorous, the authors should add a detailed theoretical or simulation model in the Main text or Supplementary Information to quantify the dynamic response between rapid polarization switching and the robot's moment of inertia. Clearly differentiating the application scope of these two control modes and exploring how to achieve control of any complex trajectory using only a single polarization state (such as a rotatable linear polarization) by optimizing the motor design or control algorithm would greatly enhance the system's universality and appeal.

The paper primarily attributes the bacterial aggregation effect to thermophoresis and explicitly rules out the influence of optical trapping forces and convection. The authors estimate the bacterial Soret coefficient ST to be $-2.45 K^{-1}$ by measuring the equilibrium diameter of bacterial clusters at different laser intensities. This conclusion provides valuable quantitative data for understanding the application of photothermal effects in biological manipulation. However, the paper does not provide sufficient experimental or simulation data to fully support its claim of "exclusivity."

First, when multiple bacteria aggregate to form a micron-sized cluster, the cluster itself could act as a new optical scatterer or resonator, generating new optical forces in the near-field region. Although the optical gradient force on a single bacterium might be weak, the clustering effect could change the force distribution. Second, while the authors use finite-element simulations to rule out macroscopic convection, local fluid dynamics on the microscale (e.g., micro-convection caused by non-uniform temperature gradients) may still play a role in the initial capture and assembly process of bacteria. To enhance the persuasiveness of this conclusion, it is recommended that the authors:

1. Provide more detailed finite-element simulation results in the Supplementary Information, showing the light field distribution near the plasmonic motor and the corresponding optical and thermophoretic force distributions in a multi-bacterial aggregation state. This would more intuitively demonstrate the dominant role of thermophoresis in long-range capture.

2. Conduct a more detailed analysis of the estimated Soret coefficient ST , comparing it with similar values from published literature and discussing the experimental uncertainties that might affect this value. For example, the non-spherical geometry of bacterial clusters and size differences between different bacterial types (e.g., *E. coli* and *S. carnosus*) could introduce estimation errors.

The paper presents the movement of both the microrobot and nanorobot along rectangular trajectories and provides curves showing velocity as a function of laser intensity. It is noteworthy that the microrobot's instantaneous velocity accelerates in the middle of the path and decelerates at the ends, while the nanorobot exhibits a more square-wave-like pattern. The authors attribute this phenomenon to local intensity variations of the Gaussian beam spot. This is a very important observation that merits a deeper quantitative analysis. In steady-state motion, the robot's optical propulsion force is balanced by the viscous drag force. However, in a non-uniform Gaussian light field, the robot's equation of motion should account for the position-dependent propulsion force $F_{\text{thrust}}(l)$ and the optical gradient force $F_{\text{gradient}}(l, x)$. By establishing a more detailed physical model and fitting the experimental data to the theoretical model (which includes the Gaussian beam intensity profile), the physical parameters of the robot's interaction with the fluid environment could be extracted more accurately, thereby enhancing the persuasiveness of the conclusions.

Additionally, the paper demonstrates that the robot can maintain maneuverability while carrying a large bacterial load, which can weigh hundreds of times its own mass. This is an impressive phenomenon, but a corresponding quantitative analysis is lacking. The formation of a bacterial cluster significantly increases the robot's effective volume and mass, which in turn changes its viscous drag coefficient in the fluid. It is recommended that the authors provide quantitative data on the robot's motion performance under different bacterial loads in the Supplementary Information and discuss the relationship between the viscous drag coefficient and the load. This is crucial for evaluating its practical utility as a "nanoscale cleaner."

Language Optimization: In the abstract, the phrase "eliminates the need for beam steering and focusing, thereby reducing system complexity and photodamage" could be made more precise by replacing "reducing" with "mitigating," which better reflects the localized nature of the photothermal effect. Furthermore, the phrasing "The two-fold orientational locking degeneracy is lifted via spin-momentum transfer by circularly polarized light pulses" is somewhat awkward and could be simplified to "Circularly polarized light pulses are used to lift the two-fold orientational locking degeneracy via spin-momentum transfer." enters the correct stable alignment state each time.

Figure Consistency: The y-axis label "Reduced orientation (Degree)" in Fig. 2e requires a clearer definition to prevent ambiguity for the reader. The SEM images in Fig. 1a and Fig. 3a could adopt a more consistent style or labeling.

Information Density: Some key experimental parameters (e.g., specific laser power, spot diameter, and EOM parameters used in different experiments) are scattered throughout the text, making it inconvenient for readers to find and replicate the experiments. It is suggested that the authors compile these key parameters into a single table and place it in the Methods or Supplementary Information section.

Version 1:

Reviewer comments:

Reviewer #1

(Remarks to the Author)

I have reviewed the revised manuscript and the detailed response to my previous comments. The authors have effectively addressed all the questions and concerns raised in the initial round of review. I specifically appreciate the addition of the supplementary estimations and calculations, which provide the necessary rigor and clarity to support the study's conclusions. The overall quality of the manuscript has been significantly improved through these revisions.

In my view, the paper now meets the high standards of Nature Communications, and I am pleased to recommend it for publication in its current form.

Reviewer #2

(Remarks to the Author)

The manuscript entitled "A nanoscale robotic cleaner" presents a highly innovative light-driven nanorobot system. By employing a single plasmonic directional antenna, the authors achieve efficient propulsion and precise orientation control under an unfocused laser field. The work ingeniously integrates photon-recoil thrust with self-alignment torque induced by linear polarization, successfully miniaturizing the robot to the sub-micron scale (diameter $< 1 \mu\text{m}$)—a 1000-fold reduction in mass compared to state-of-the-art light-driven metavehicles. The system demonstrates the ability to trace complex trajectories and execute sophisticated biological tasks, such as the capture, transport, and reversible assembly of bacteria. The authors have provided comprehensive and persuasive responses to all concerns raised in the initial review, particularly regarding physical mechanisms, quantitative benchmarks, 3D stability, and data transparency.

The in-depth analysis of the "self-alignment" dynamics successfully justifies the use of auxiliary pulses to overcome orientational metastability and Brownian noise during turns.

The theoretical derivations for 3D stability and the detailed force analysis (thermophoresis vs. optical trapping) significantly enhance the robustness of the conclusions.

The supplementary experimental parameter summary table in SI consolidates all core data pertinent to the experiment, encompassing voltage, power density, and spot size, offering readers a clear reference. The authors have fully addressed all major technical points. The revised manuscript represents a significant advance in the field of light-driven nanorobotics, characterized by its conceptual novelty, physical depth, and experimental excellence. I recommend the paper for publication after minor editorial adjustments.

Reply to reviewer comments on Nature Communications Manuscript NCOMMS-25-56045-T

We thank the reviewers for the careful and positive evaluation of our work as well as for their valuable remarks. We have addressed all points in detail below. Direct citations of the reports are printed in black, our replies in blue font. Modifications are also visible in a marked-up version of the manuscript. Page numbers in our reply refer to the revised, marked-up version. Changes to the manuscript and SI are summarized in **blue bold**.

Reviewer #1 (Remarks to the Author):

The manuscript presents a light-driven micro/nanorobot, miniaturized to sub-micrometer dimensions, that functions as a robotic cleaner for biological applications. The robot is powered and steered by a single-wavelength unfocused laser beam using a simplified control architecture where two sets of plasmonic antennas provide both propulsion and orientation control. The robot's direction of motion is locked perpendicular to the driving linear polarization, while deterministic turns are executed using short pulses of circularly polarized light. Leveraging a combination of optical trapping and long-range thermophoretic forces, the robot can efficiently capture and assemble large clusters of bacteria, which can then be released at designated locations. The work is supported by FEA simulations, along with experimental validation demonstrating high-speed maneuvering up to 50 $\mu\text{m/s}$.

The concept of a sub-micron, light-driven robotic cleaner is compelling, and the simplified, miniaturized design is a smart approach. However, the manuscript's impact could be substantially improved. The primary weakness is the lack of a direct comparison to the authors' own previously reported chiral antenna microdrones (refs 1, 13) to clarify the pros and cons of this new design. Additionally, the manuscript needs a more rigorous analysis of the trapping mechanism and a quantitative assessment of the robot's maneuverability and cleaning efficiency. The work requires major revisions to address these critical questions.

Major Comments:

1. The manuscript does not provide a clear comparison to the authors' highly relevant prior work on chiral antenna microdrones. While the new design is clearly smaller, a detailed discussion of the trade-offs is missing. The authors should provide an explicit comparison of the pros and cons of this new "car-like" design versus the previous "drone-like" one, clarifying if the smaller size is a direct consequence of the simplified control scheme and how the propulsion efficiency compares quantitatively to the multi-motor chiral designs.

We thank the reviewer for this insightful comment. We agree that a clear comparison with our prior chiral microdrone design is important, and **we have now included a detailed**

discussion in the revised manuscript (Page 11, Line 419-432) and supplementary information (S3).

Both the previous “drone-like” microdrone and the present “car-like” nanorobot were developed to address the fundamental challenge of achieving controlled propulsion in fluid environments, where Brownian motion perturbs both translation and orientation. Thus, orientation control is essential in addition to unidirectional thrust.

In the *drone-like* system, we focused more on higher maneuverability, achieved via circularly polarized light fields with opposite helicities which generate counteracting optical torques to balance orientation during propulsion. In contrast, the *car-like* design is more focused on high propulsion efficiency. It employs a single plasmonic directional antenna illuminated by a linearly polarized light field. This configuration provides both unidirectional photon-recoil thrust and a passive torque trap that aligns the robot’s orientation with respect to the polarization axis, eliminating the need for multi-beam torque balancing.

Below, we summarize the key advantages and trade-offs of both approaches:

Advantages of the *car-like* nanorobot:

- Miniaturization: The single-antenna design enables size reduction below 1 μm .
- Simplified control: Only a single unfocused light field is required; no beam splitting or polarization balancing is needed.
- Higher propulsion efficiency: Because orientation is self-stabilized, the complete photon momentum and not only a projection thereof contributes to forward thrust. As a result, the translational velocity is significantly higher for the same optical power.

Advantages of the *drone-like* microdrone:

- Higher maneuverability: Independent control of recoil by two motors provides access to all degrees of freedom in a 2D plane, without changing the robot’s body orientation. Furthermore, complex swirling motions are possible by only activating one of the four motors. However, this requires precise alignment of two beams, power balancing, and a finite spatial separation of antenna elements, which limits device miniaturization and increases optical system complexity.

Trade-offs of the *drone-like* and the *car-like* nanorobot design:

- The *drone-like* design provides full in-plane motion freedom but requires two-beam control and cannot be efficiently scaled down.
- The *car-like* design sacrifices rotational degrees of freedom but gains sub-micron device size, simpler one-beam operation, and improved propulsion efficiency.

To quantitatively compare propulsion efficiency, we now include direct force simulations and efficiency metrics in Figure R1. Under equal total optical power ($0.6 \text{ mW}/\mu\text{m}^2$):

- The *drone-like* design generates $\sim 0.12 \text{ pN}$ net propulsive force.
- The *car-like* design generates $\sim 0.30 \text{ pN}$, a $2.5\times$ increase.

We further define propulsion efficiency η as velocity per optical power. The *drone-like* system yields $\eta \approx 13.3 \text{ } (\mu\text{m}\cdot\text{s}^{-1})/(\text{mW}/\mu\text{m}^2)$, while the *car-like* system reaches $\eta \approx 46.9 \text{ } (\mu\text{m}\cdot\text{s}^{-1})/(\text{mW}/\mu\text{m}^2)$ for the microrobot and $\eta \approx 121.2 \text{ } (\mu\text{m}\cdot\text{s}^{-1})/(\text{mW}/\mu\text{m}^2)$ for the nanorobot.

Figure R1 Illustration of two types of light driven devices. **a.** “drone-like” device. The orange(red) solid line represents the force direction generated from 830nm(980nm) laser. The total force used for propelling the device is indicated by the dashed lines. **b.** “car-like” device.

The table below provides a comparison of the two designs:

Feature	“Drone-like” Design	This Work: “Car-like” design
Typical Size (Diameter)	~2.5 μm	920nm (single motor) 1.5 μm (four motor)
Propulsion Mechanism	Photon recoil from chiral plasmonic antennas	Photon recoil from directional plasmonic antenna + Orientation locking
Control Inputs	Two independent circularly polarized beams	One unfocused beam; polarization sequencing (linear + circular)
Degrees of Freedom	All degrees of freedom in 2D translation + rotation	Translation with polarization-locked orientation (rotation direction controlled via CP pulses)
Propulsion Efficiency (velocity per optical power)	$\sim 13.3 \frac{\mu\text{m/s}}{\text{mW}/\mu\text{m}^2}$	$46.9 \frac{\mu\text{m/s}}{\text{mW}/\mu\text{m}^2}$ (four motor) $121.2 \frac{\mu\text{m/s}}{\text{mW}/\mu\text{m}^2}$ (single motor)
Typical Speed	~8 $\mu\text{m/s}$	15 $\mu\text{m/s}$ (four motor) ~40 $\mu\text{m/s}$ (single motor)
Optical Setup	Requires beam splitting, alignment, and power balancing	Single beam
Primary Advantage	Full maneuverability in 2D plane	Miniaturization + enhanced propulsion efficiency + simplified setup
Primary Trade-off	More complex setup	Reduced degrees of freedom (addressed via polarization sequencing)

We thank the reviewer again for prompting this clearer and more quantitative comparison.

2. The manuscript also focuses exclusively on 2D motion and trapping but fails to address the robot's stability in the third dimension. While the self-correcting torque prevents in-plane rotation, there is no discussion of what prevents the disc-shaped robot from tilting or tumbling out of the xy-plane, especially during high-speed motion and turns. The authors must clarify the mechanism responsible for maintaining out-of-plane stability, such as the role of vertical radiation pressure or hydrodynamic forces.

We thank the reviewer for raising this important point. Although the manuscript focused on in-plane motion demonstration, the discussion about how to stabilize the motion in out-of-plane is indeed essential. **We have expanded this discussion in the revised manuscript (Page 5, Line 139-141) and refer to Supplementary Information S10.1 for additional details on the motion dynamics.**

In our experiment, the micro- and nanorobot operates in close proximity to the substrate surface (about several hundreds of nanometers), where the downward optical radiation force is balanced by the electrostatic repulsive force from the charged substrate. Taking the four-motor design as an example (Figure 2), the out-of-plane optical force F_{oz} was simulated in COMSOL using Maxwell's stress tensor. We found downward optical force $F_{oz} = -1.35$ pN for a laser intensity of $0.3 \text{ mW}/\mu\text{m}^2$, mainly originating from optical radiation pressure.

The static repulsive force can be approximated as the interaction between an infinite charged plate (the substrate, with charge density σ_1 and radius R) and a finite charged microscale disk (diameter $1.5 \mu\text{m}$, charge density σ_2 , spacing distance z). The electric field from the infinite plate is

$$E_z = \lim_{R \rightarrow \infty} \frac{\sigma_1}{2\epsilon} \left(1 - \frac{z}{\sqrt{z^2 + R^2}}\right) = \frac{\sigma_1}{2\epsilon}$$

where ϵ is the permittivity of water. The resulting electrostatic repulsive force acting on the disk is

$$F_s = \frac{\sigma_1 \sigma_2 \pi r^2}{2\epsilon}$$

where r is the radius of microscale disk.

Assuming the substrate and disk carry similar surface charge densities ($\sigma_1 = \sigma_2$), and equating F_s with F_o , we estimate a surface charge density of $\sigma = 3.3 \times 10^{-5} \text{ C/m}^2$, corresponding to roughly 360 elementary charges on the microscale disk.

In realistic aqueous environments with finite ionic strength, the electrostatic interaction is screened, making the force distance-dependent:

$$F_s(d) = \frac{\sigma_1 \sigma_2 \pi r^2}{2\epsilon} e^{-d/\lambda_D}$$

where λ_D is the Debye length. In pure water solution, λ_D is roughly $1 \mu\text{m}$. This screening allows the equilibrium position to self-adjust under different optical intensities, maintaining balance between optical radiation pressure and electrostatic repulsion.

Moreover, the out-of-plane optical force is approximately 7.5 times stronger than the in-plane thrust, providing a strong restoring component that prevents tilting or tumbling even during high-speed motion or turns. Hydrodynamic damping in the viscous medium further suppresses any vertical perturbations. Small out-of-plane velocities are quickly dissipated

by viscous drag, ensuring the nanorobot maintains a stable orientation parallel to the substrate.

Together, these optical, electrostatic, and hydrodynamic effects stabilize the robot in the z-direction, allowing it to remain stably aligned within the xy-plane while being precisely steered laterally.

3. The analysis of the trapping mechanism is insufficient for a dynamic "cleaner." The paper establishes an equilibrium cluster based on a static ~ 1 kBT potential, but this does not account for the hydrodynamic drag force that trapped bacteria will experience during motion. A deeper potential well is required to retain the cluster, and the manuscript needs a thorough discussion of assembly performance and stability under dynamic conditions. The trapping is also a 3D process, as the robot collects bacteria from different heights, yet the potential analysis in Figure 4d is limited to the 2D xy-plane. The authors must provide a cross-sectional (e.g., xz-plane) analysis of the thermophoretic trapping potential to properly characterize the 3D trapping volume and strength.

We thank the reviewer for these insightful comments and would like first to clarify a potential misunderstanding. The trapped bacterial cluster is not confined by a potential depth of only $1 k_B T$; rather, as shown in Fig.~4d, the thermophoretic trapping potential is substantially deeper. The $1 k_B T$ contour is introduced solely to estimate the effective trapping region, as it represents an equilibrium boundary where the thermophoretic force is balanced by Brownian motion. We agree with the reviewer that, for a moving "cleaner," both the hydrodynamic drag acting on the trapped bacteria and the three-dimensional trapping landscape are important factors. In the revised manuscript, we have expanded the analysis to explicitly account for these effects.

1. Hydrodynamic drag force on trapped bacteria

We now include a quantitative estimate of the hydrodynamic drag force acting on trapped bacteria during robot motion in revised manuscript (Page 9-10, Line 349-364) and Supplementary Information S8.3.

Taking carnosus bacteria (diameter of roughly $1 \mu\text{m}$) as examples, the hydrodynamic drag force, which acting on the bacterium carried by a moving microrobot, can be expressed as $F_d = 6\pi\eta r v$, where η is water viscosity, r , v are the radius and velocity of moving bacterium respectively. As we see in Supplementary Video 5, when a large bunch of bacteria is assembled around the microrobot, the typical moving velocity is quite low (below $0.5 \mu\text{m/s}$). In this case, the hydrodynamic drag force can be estimated as 0.0019 pN , which is far below the thermophoretic trapping force, which is estimated by $F_t = k_B T S_T \nabla T$. Taking the value of $\nabla T = 10^7 \text{ K/m}$ (referred from Figure S5), F_t is 0.1 pN , which is roughly 50 times larger than the drag force. But the existence of this drag force still will distort the trapping potential as shown in Figure R2, where we tune the moving velocity from 0 nm/s to 500 nm/s . As we can see from the inset, when the microrobot moves left with trapped bacteria, for the left-most trapped bacterium, the drag force effectively will enhance the trap performance. In the contrary, for the right-most bacterium, the drag force will counteract the thermophoretic force, leading to a decrease of the effective trapping potential.

Figure R2 Illustration of trapping potential with influence of hydrodynamic force. Inset: Sketch of the force distribution when the robot moves towards the left indicated by the black arrow labeled v . Drag force, F_d , and trapping force, F_t , add up or cancel.

In Supplementary Video 5 (time frames 17s-22s), when the microrobot accelerates, due to the inertia, the previously captured bacterium is detached from the device, due to the increasing hydrodynamic drag force.

In summary, thermophoretic forces indeed contribute to the bacterial trapping mechanism. Importantly, this effect is self-stabilizing: as the robot's translational velocity changes, the number of effectively trapped bacteria adjusts automatically.

2. Thermophoretic trapping in xz plane

We now include the 3D trapping in the revised manuscript (Page 9, Line 336-340) and supplementary information S8.2.

The thermophoretic trapping potential in the xz-plane was obtained from the simulated temperature distribution shown in Fig. R3a. Using the Soret coefficient estimated from our measurements, we calculated the corresponding thermophoretic potential landscape (Fig. R3b). This analysis shows that trapping is not restricted to the 2D xy-plane but extends into the full 3D space surrounding the robot. In particular, the thermophoretic force exceeds the hydrodynamic drag over a broad region above and around the robot, pulling approaching bacteria toward the device from all directions. The resulting equipotential surfaces form a semi-hemispherical confinement region, consistent with a true 3D trapping volume.

Due to limitations of optical imaging, bacteria trapped directly above the microrobot or nanorobot cannot always be clearly resolved. Nonetheless, this behavior is observed experimentally: during the collection process, not all bacteria accumulate solely at the rim of the device—some are captured and stacked on the top surface, confirming the presence of out-of-plane thermophoretic confinement.

Figure R3. Thermophoretic trapping behaviour in xz plane. a, temperature distribution around the four-motor design when averaged laser power is $0.28 \text{ mW}/\mu\text{m}^2$. b, The calculated Thermophoretic trapping potential with assigned Soret coefficients of 2.45 K^{-1} . c, A linecut of trapping potential in b when $x=0$.

4. The claims about the robot's performance are not sufficiently supported by quantitative data. The assertion that the robot "remains fully maneuverable" after collecting a large bacterial cluster is a significant overstatement. The increased hydrodynamic drag must impact performance, and the authors need to provide a quantitative analysis comparing key metrics like maximum speed and turning precision before and after bacterial assembly. It is also unclear how active motion improves cleaning efficiency over a static trap with a long capture range. A clear rationale, ideally supported by data comparing the clearance rates of a mobile versus a static robot, is needed to justify the "cleaner" concept.

We thank the reviewer for this constructive comment. We admit that the "fully maneuverable" statement can be misleading and appear overstated, especially when a bunch of bacteria is captured around the robotic device. **Hence, we would remove the word of "fully"**. Our main claim is that even when the robot is loaded with a large number of bacteria and its translational velocity is reduced, its orientation can still be reliably controlled. We also agree that clarifying the robot's performance before and after bacterial collection, especially **include the discussion of the influence of hydrodynamic force will further strengthen our manuscript. We applied corresponding revisions in main text (Page 9-10, Line 349-364) and supplementary information S9.1 and S9.2.**

1. Maximum speed after bacteria assembly

Before loading the bacteria cluster, the micro- or nanorobot can be maneuvered with a very fast velocity. However, it's not possible to trap bacteria with such high moving velocity. As we discussed before (the third point for Reviewer 1), the hydrodynamic force acting on the bacteria will hinder the thermophoretic trapping force, which will effectively decrease the trapping range. Taking the four motor design as an example, when the laser intensity is $0.3 \text{ mW}/\mu\text{m}^2$, we obtain an optical force up to 0.15 pN , which should be balanced with the drag forces acting on the robotic device and trapped bacteria. The total drag force acting on robotic device can be simulated via computational fluid dynamics (CFD, COMSOL). Assuming a fixed height above the substrate, the total drag force increases linearly with the velocity, as shown in Figure R4a.

To simplify the model, we assume that the trapped bacteria are spherical with radius r . When the microrobot carries a bacterial cluster, the total hydrodynamic drag force F_{drag} is balanced by the optical thrust generated by the motor design:

$$F_o = F_{\text{drag}} = N \times 6\pi\eta r v + Dv,$$

where D is the drag coefficient extracted from CFD simulations as ($D=1.3\text{e-}8\text{N}/(\text{m/s})$), N is the number of loaded bacteria, η is the viscosity of water, and v is the microrobot velocity. The term $6\pi\eta r v$ represents the hydrodynamic drag force acting on a single spherical bacterium.

Assuming the robotic device moves with a velocity of $5 \mu\text{m/s}$ and the average radius of bacteria is roughly 500nm , then the maximum number of trapped bacteria should be less

than $N \leq \frac{F_o - D}{6\pi\eta r v} = 2.1$. This also explains that in Supplementary Video 6, the typical moving velocity is quite low. From Figure R4b, we find that, when the moving velocity exceeds $7.5 \mu\text{m/s}$, limited by the drag force, the number of effective trapped bacteria will decrease below one, which indicates that the optical thrust force is not enough to balance the total drag force acting on the robots with trapped bacteria. Another aspect, as discussed in #3, the thermophoretic force around the rim of robotic device can reach up to 0.1pN , which is enough to capture the bacteria. This also explains the trapping behavior is most likely around the rim of device.

Figure R4 **a**, The drag force when the microrobot (nanorobot) moves with different translational velocity. **b**, Maximum trapped bacteria with different translational velocities with a fixed optical thrust force of 0.15pN . The blue dashed line indicates the cutoff velocity which can be used for bacteria trapping.

2. Turning precision after bacteria assembly

The turning behavior is not affected by the presence of trapped bacteria. As shown in Supplementary Video 5, even after the microrobot has assembled a large number of bacteria, its orientation still follows the linear polarization reliably. This is because the bacteria are not rigidly attached to the robot's surface; instead, they remain slightly separated due to electrostatic repulsion via surface charges. Consequently, even when the bacterial cluster undergoes random rotational fluctuations driven by Brownian motion, the robot's orientation is not significantly influenced. The orientation dynamics remain dominated by the intrinsic optical self-locking mechanism of the microrobot.

3. Cleaning efficiency over a static or mobile microrobots

We agree that a static trap can attract bacteria within its thermophoretic capture range. However, its cleaning capability is fundamentally limited by diffusion: bacteria outside the ~5–10 μm capture radius approach slowly, leading to long clearance times.

To clarify the advantages of active motion, we now include a simple model comparing clearance rates:

In a static trap, in time t , it will mostly clean a single volume of $2\pi R_c^3/3$ around itself. Beyond that, bacteria only reach it by slow Brownian diffusion.

In contrast, an actively propelled robot transports this spherical capture volume through space. As it moves with speed v , it effectively sweeps out a much larger “cleaned” volume per unit time (V_{mobile}), of order

$$\frac{dV_{mobile}}{dt} = \frac{\pi R_c^2 v}{2},$$

where R_c is the radius of static trapping. Over experimentally relevant timescales, this swept volume exceeds the static capture volume by one to two orders of magnitude, meaning the mobile robot encounters and removes bacteria from many different regions instead of repeatedly interacting with the same local neighborhood. This is consistent with our time-lapse observations, where mobile robots visibly clear extended areas, whereas static hot spots only accumulate bacteria in a confined region.

We have added the 3D “swept-volume” argument to the revised manuscript (Page 10, Line 374-377) and supplementary information S9.3 to quantitatively justify the “cleaner” designation.

5. All demonstrations show a single robot. For practical applications like large-area cleaning, the authors should discuss the feasibility of collective operation. Potential limitation should be acknowledged.

We thank the reviewer for the constructive comment. While our manuscript focuses on the capabilities of a single robot, the underlying propulsion and control mechanisms are compatible with multi-robot operation, and **we now clarify this in the revised manuscript (Page 11, Line 412-418)**. At the same time, we add some discussion about the limitations.

1. Collective operation for multiple devices.

Collective operation is feasible using spatial light modulators (SLMs) or digital micromirror devices (DMDs), which can divide a single illumination field into multiple independently controlled sub-fields. Each sub-field can be assigned to an individual polarization state and intensity profile, enabling simultaneous control of several robots in parallel. In this scheme, each robot behaves as a local agent responding to its own optical “control envelope,” allowing scalable parallelized operation for large-area cleaning.

2. Possible limitations.

Despite the general feasibility, several limitations must be acknowledged:

(a) Optical power distribution

Dividing a laser into many sub-fields may reduce the power available per robot. Since propulsion speed increases approximately linearly with intensity, large collectives may require proportionally more total laser power to maintain individual performance.

(b) Field uniformity and alignment

For multi-robot systems, each sub-field must maintain:

- uniform intensity across the robot's working region,
- stable polarization states,

Optical imperfections may lead to unequal propulsion forces, drift, or loss of orientation control.

(c) Control bandwidth and synchronization

SLM/DMD refresh rates limit how quickly:

- steering commands can be updated,
- trajectories can be adjusted for each robot,
- feedback control can be implemented.

This places a practical upper limit for real-time coordinated operations.

To summarize, we now acknowledge that although collective operation is feasible using SLM-based multiplexing of polarization and intensity, several practical limitations must be considered. These include (i) reduced per-robot optical power as the laser field is subdivided, (ii) the need for highly uniform and stable optical fields over large areas, (iii) the need of high control bandwidth to dynamically implement feedback system. **We have incorporated these points in the revised Discussion section (Page 11, Line 412-418).** Together, these considerations outline both the opportunities and the challenges associated with scaling our system for multi-robot cleaning applications.

Minor Comments

1. The overall readability is hindered by the manuscript's structure. The captions for Figures 2, 3, and 4 are excessively long and contain methodological details and interpretations that belong in the main text, disrupting the narrative flow. It is strongly recommended that the authors move these details to the main body of the text and revise the captions to be concise descriptions of the data.

Thanks for this suggestion. **We have shortened the figure captions in the revisions and moved some details to the main text.**

2. Several figures also lack the necessary clarity for straightforward interpretation. The rendered schematic in Figure 1a is difficult to read and should have a more transparent body to show the internal antennas. The caption for Figure 1b should state the time interval between the superimposed frames to provide a sense of speed. Due to the motor's high symmetry, a coordinate system or an arrow defining the "+y" direction of propulsion must be added to Figures 2a and 2b to resolve ambiguity.

We are grateful for the reviewer's suggestion. **Now Figure 1a has been updated with a more transparent body and more distinguishable structures.**

The time interval between superimposed frames in Figure 1b has been added as presented in Supplementary Figure S1.

A coordinate system has been added in Figure 2 and Figure 3.

3. Some important technical details are also missing from the main text. For instance, the dimensions of the dimer for the orienting torque cannot be found in the main manuscript and should be included for completeness.

We have added the dimensions of the dimer for orientation effects in the revised manuscript (Page 4, Line 111) and Fig. S4.

Reviewer #2 (Remarks to the Author):

The paper proposes a mechanism that uses a restoring torque induced by linear polarized light to achieve orientational locking, which stably aligns the robot along the linear polarization axis to maximize propulsion efficiency. I think some issues need to be answered or solved before considering the publication of this submission.

The authors state that a brief circularly polarized light pulse is required to achieve precise 90° turns. The need for this auxiliary pulse seems to contradict, to some extent, the core advantage of the "self-alignment" mechanism, which is to control direction using only linear polarized light.

From a deeper physical perspective, this phenomenon reflects the dynamic relationship between the optical torque and the robot's intrinsic angular momentum and inertia. When the polarization state is switched rapidly (approx. 1 ms), the robot's moment of inertia may prevent it from immediately capturing the new stable alignment direction, instead causing it to settle in a metastable maximum of the orientational potential well. In this state, any minor disturbance, such as Brownian motion, could cause the robot to rotate randomly to the left or right, leading to a loss of deterministic control over the turning direction. The authors use a circularly polarized light pulse to provide an additional, deterministic torque via spin-momentum transfer, overcoming this uncertainty and ensuring the robot enters the correct stable alignment state each time.

Although this turning mechanism is effective, its potential limitation lies in controlling more complex and smooth trajectories. The paper mentions in the discussion that "smooth paths can be implemented just by turning the linear polarization," which implies two different control modes: one for straight motion and slow turns using pure linear polarization control, and another for precise 90° turns using a hybrid of linear and circular polarization. To make the paper more rigorous, the authors should add a detailed theoretical or simulation model in the Main text or Supplementary Information to quantify the dynamic response between rapid polarization switching and the robot's moment of inertia. Clearly differentiating the application scope of these two control modes and exploring how to achieve control of any complex trajectory using only a single polarization state (such as a rotatable linear polarization) by optimizing the motor design or control algorithm would greatly enhance the system's universality and appeal.

We thank the reviewer for this thoughtful and insightful comment. As suggested, we now provide a theoretical model that clarifies the distinction between the two control modes—(i) smooth trajectory control using only slowly rotating linear polarization and (ii) deterministic rapid turning assisted by a brief circularly polarized pulse. A discussion of **this model and its implications have been added to the revised manuscript (Page 12, Line 402-408) and Supplementary Information S7.**

To analyze the ability of the robot to follow a rotating polarization state, we consider the optical torque generated by the self-alignment mechanism. For the four-motor microrobot (Fig. 2b), the optical torque can be expressed as:

$$\tau_{opt} = -\tau_0 \sin(2\theta)$$

where θ is the robot's orientation relative to the linear polarization. When the polarization direction is rotated at a constant angular speed ω , the robot experiences both this optical restoring torque and a viscous drag torque. In the low-Reynolds-number regime, the drag torque increases linearly with the rotational velocity ω_r , yielding:

$$\tau_{drag} = k\omega_r$$

where the rotational drag coefficient k is obtained from CFD simulations. For the robot geometry and a robot–substrate spacing of 500 nm, the simulated relationship between drag torque and angular velocity is shown in Fig. R5a.

Transforming to the frame rotating with the polarization angle $\phi(t) = \omega t$, the relative angle θ obeys:

$$\dot{\theta} = \omega - \int_0^t \omega_r d\omega_r$$

Balancing optical torque with drag torque leads to the rotational dynamics:

$$\dot{\theta} = \omega - \frac{\tau_0}{k} \sin(2\theta)$$

In a steady state ($\dot{\theta} = 0$), we obtain:

$$\sin(2\theta^*) = \frac{\omega}{\tau_0/k} = \frac{\omega}{\Omega}$$

A real, stable solution exists only when $\omega < \Omega$. In this regime, the robot maintains a constant phase lag relative to the rotating polarization and follows the linear polarization smoothly. When $\omega > \Omega$, no steady-state solution exists, and the orientation angle continually drifts—i.e., the robot cannot keep up with the polarization rotation and exhibits “slip.”

Figures R5b and R5c show numerical solutions in both regimes. At a moderate rotation rate (e.g., $\omega = 2 \text{ rad/s}$), the robot tracks the polarization closely, whereas at higher rotation speeds the orientation fails to follow and slips repeatedly.

In this simplified model, we neglect Brownian rotational noise. However, this noise becomes important during rapid 90° polarization changes: when the optical potential is at metastable position, thermal fluctuations can push the robot into a stable region, causing the turning direction (left vs. right) to become nondeterministic.

Figure R5. **a** Relationship between drag torque and angular velocity obtained from CFD simulations. **b** Numerical solution of the rotational dynamics when the polarization

rotation speed ω is smaller than the locking rate Ω , showing stable phase locking. **c** Evolution of the polarization angle and the corresponding robot orientation in the regime $\omega > \Omega$, demonstrating that the robot can't faithfully follow the rotating polarization.

When a short circularly polarized light pulse is applied, the robot experiences a unidirectional optical torque, resulting in deterministic rotation at an angular velocity of approximately 11.8 rad/s (as derived from Fig. R5). A pulse duration of ~100 ms therefore produces a rotation of about 67.7° in the ideal, noise-free case—sufficient to reliably nudge the robot to rotate into the correct direction.

From the analysis above, both methods can be used to tune the robot's orientation. However, applying short pulses of circularly polarized light is more suitable for rapid reorientation necessary under high-speed operation. As discussed earlier, the rotation speed of the linear polarization is inherently limited; if it is too fast, the device's orientation cannot follow, leading to orientation slip. This consideration underlies our demonstration design: because the robot operates at relatively high velocities, prompt and reliable orientation changes are essential. Besides, rapid polarization switching combined with spin angular momentum transfer provides a deterministic way of executing sharp turns, thereby avoiding large-radius curves and enabling precise cleaning operations.

As discussed above, continuous rotation of the linear polarization would allow smooth trajectory control entirely through the self-alignment mechanism, provided the rotation speed remains below the locking rate Ω . This capability could be realized in future experiments by integrating a motorized rotating polarizer for continuous polarization rotation.

The paper primarily attributes the bacterial aggregation effect to thermophoresis and explicitly rules out the influence of optical trapping forces and convection. The authors estimate the bacterial Soret coefficient ST to be -2.45 K^{-1} by measuring the equilibrium diameter of bacterial clusters at different laser intensities. This conclusion provides valuable quantitative data for understanding the application of photothermal effects in biological manipulation. However, the paper does not provide sufficient experimental or simulation data to fully support its claim of "exclusivity."

First, when multiple bacteria aggregate to form a micron-sized cluster, the cluster itself could act as a new optical scatterer or resonator, generating new optical forces in the near-field region. Although the optical gradient force on a single bacterium might be weak, the clustering effect could change the force distribution. Second, while the authors use finite-element simulations to rule out macroscopic convection, local fluid dynamics on the microscale (e.g., micro-convection caused by non-uniform temperature gradients) may still play a role in the initial capture and assembly process of bacteria. To enhance the persuasiveness of this conclusion, it is recommended that the authors:

1. Provide more detailed finite-element simulation results in the Supplementary Information, showing the light field distribution near the plasmonic motor and the corresponding optical and thermophoretic force distributions in a multi-bacterial

aggregation state. This would more intuitively demonstrate the dominant role of thermophoresis in long-range capture.

We thank the reviewer for this thoughtful and constructive comment. We agree that additional clarification is needed regarding the potential contributions of optical forces and microscale convection.

In Fig. R6, we provide a direct comparison between the simulated optical forces and the thermophoretic forces acting on bacteria near the robot. For simplicity, we model each bacterium as a sphere of 1 μm diameter with a refractive index of 1.38 [*Nano Lett.* 2009, 9, 10, 3387–3391]. The optical trapping force on each bacterium is obtained by integrating Maxwell's stress tensor over its surface. As shown in Fig. R6a, the optical force is on the order of a few femtonewtons at an excitation intensity of $0.3 \text{ mW}/\mu\text{m}^2$, and its direction does not consistently point toward the robot center. This behavior arises because the plasmonic motors are designed to scatter light asymmetrically to generate photon recoil, resulting in a highly nonuniform near-field around the robot; combined with the fact that the bacterial refractive index is close to that of water, the optical gradient force is far too weak to drive cluster formation.

In contrast, the thermophoretic force, originating from the temperature gradient around the robot, always points from the colder fluid toward the hotter region near the plasmonic motor. As shown in Fig. R6b, its magnitude is orders of magnitude larger than the optical gradient force, and it consistently directs bacteria toward the robot center, explaining the observed long-range and robust assembly behavior.

Figure R6. Comparison of optical and thermophoretic forces acting on bacteria near the robot. **a** Near field electric-field distribution (logarithmic scale) under x-polarized excitation, with red arrows indicating the optical force vectors (both directions and amplitudes) on individual bacteria. **b** Temperature distribution under the same illumination, with white arrows indicating the magnitude and direction of the resulting thermophoretic forces.

Microscale convection is discussed in Supplementary Note S11. The simulated flow velocities are below 10 nm/s, corresponding to forces of only $\sim 0.02 \text{ fN}$ —negligible compared with thermophoretic drift. We agree that such weak flows may help initiate the early-stage motion of nearby bacteria, but they cannot account for the full 3D assembly process. In fact, because convection circulates tangentially around the heated region, it

would oppose the trapping of bacteria located above the robot surface. In contrast, we observe that the assembly occurs throughout a hemispherical trapping volume, which cannot be explained by convection alone.

We added these discussions in the revised manuscript (Page 8, Line 292-294) and to the Supplementary Information S8.1.

2. Conduct a more detailed analysis of the estimated Soret coefficient S_T , comparing it with similar values from published literature and discussing the experimental uncertainties that might affect this value. For example, the non-spherical geometry of bacterial clusters and size differences between different bacterial types (e.g., *E. coli* and *S. carnosus*) could introduce estimation errors.

We thank the reviewer for this valuable suggestion. As noted, the Soret coefficient is not a fundamental material constant; it is highly sensitive to a wide range of parameters. It depends on particle size, geometry, and surface charge, all of which differ between bacterial species and even between individual cells. The surrounding medium also plays an important role, as ionic strength, pH, and buffer composition strongly affect the electric double layer. In addition, proximity to solid surfaces can significantly modify local thermo-osmotic slip, further altering the effective thermophoretic mobility. Biological cells introduce additional complexity due to their heterogeneous surfaces and anisotropic shapes. For these reasons, the Soret coefficient obtained from our measurements should be interpreted as an effective value specific to our near-surface plasmonic heating environment.

To further validate the plausibility of our estimate, we also evaluated the theoretical Soret coefficient of bacteria. Although experimental reports of bacterial S_T are limited, a commonly used expression derived from thermodynamic and electrokinetic considerations (Annual Review of Fluid Mechanics, 1989, 21, 61–99; Langmuir, 2007, 23, 9221–9228) is

$$S_T = -\frac{2\pi R}{k_B T^2} \left[\frac{2\Lambda_l}{2\Lambda_l + \Lambda_p} \right] \left(\varepsilon + T \frac{\partial \varepsilon}{\partial T} \right) \zeta^2$$

Here, R is the bacteria radius, k_B is Boltzmann constant, and Λ_l and Λ_p are the thermal conductivities of water and bacteria, respectively. The thermal conductivity of bacteria is set to be 0.1 W/(m·K) (*Lab Chip*, 2023, 23, 2411-2420). The static permittivity of water and its temperature derivative $\partial \varepsilon / \partial T$ are obtained from (*J. Phys. Chem. Ref. Data* 19, 371–411 (1990)). The zeta potential of the solution ζ is taken to be -25mV (*Front. Microbiol.* 7:1732). Besides, the motion of bacteria close to the substrate is significantly influenced by the hydrodynamic boundary effects, which can increase the Soret factor by an enhancement factor of (*Phys. Rev. Lett.* 2016, 116 (13), 138302):

$$\Phi_H = 3(1 + H) \frac{(2 + 6H + 3H^2) \ln\left(\frac{H+1}{H}\right) - 1.5(3 + 2H)}{2 + 9H + 6H^2 - 6H(1 + H^2) \ln\left(\frac{H+1}{H}\right)}$$

where $H = h/R$ and h is the nanoparticle-surface distance.

Figure R7 Calculated effective Soret coefficient based on the theoretical model for (a) varying bacterial radii (with fixed zeta potential $\zeta = -25$ mV) and (b) varying zeta potentials (with fixed bacterial radius of 500 nm).

Figure R7 shows the resulting effective Soret coefficients for (a) different bacterial radii and (b) different zeta potentials. The values extracted from our hydrodynamic trapping model fall well within these theoretically expected ranges.

Figure R7a shows that increasing bacterial size leads to an almost linear increase in the effective S_T , explaining why different bacterial species or cell sizes exhibit different trapping ranges. Our analytical model assumes spherical bacteria, but real cells—especially *E. coli*—are anisotropic, which introduces additional uncertainty that is difficult to model precisely. Higher surface charge magnitudes (larger $|\zeta|$) also increase S_T , consistent with our observation that *S. carnosus*—which appears to carry more surface charge—aggregates more readily and forms more frequent clusters than *E. coli*.

We also note that similar magnitudes of thermophoretic forces have been inferred in related optical-thermophoretic trapping studies. For example, in Biomed. Opt. Express 12, 3917–3933 (2021), plasmonic optical fibers were used to assemble *E. coli*, where thermophoretic forces exceeding ~ 100 fN were needed to overcome hydrodynamic drag—comparable to our estimates (Reviewer 1, Comment 3) and consistent with Sci. Rep. 6, 35814 (2016). Likewise, Phys. Rev. Research 6, L032061 (2024) reported effective Soret coefficients as large as -3 K⁻¹ for *Salmonella* in dense suspensions, noting that S_T increases steeply at higher concentrations due to frequent cell–cell collisions and long-range hydrodynamic interactions.

In our experiments, the effective Soret coefficient is obtained by identifying the equilibrium position where thermophoretic drift balances Brownian motion. This estimate carries inherent uncertainty because different bacterial sizes, shapes, and surface charges produce different trapping radii, and because the assembled clusters are not perfectly

symmetric. As reflected in Fig. 4e, cluster diameters vary across experiments due to local intensity variations and biological heterogeneity.

In summary, our reported S_T should be regarded as a reasonable effective estimate, consistent with theoretical expectations and comparable experimental studies. At the same time, it provides a practical approach for rapidly estimating the effective Soret coefficient of bacterial suspensions without requiring complex electrochemical measurements.

We have added the theoretical model, expanded discussion of uncertainties, and included additional citations in the revised manuscript (Page9, Line 328-331) and Supplementary Information S9.4.

The paper presents the movement of both the microrobot and nanorobot along rectangular trajectories and provides curves showing velocity as a function of laser intensity. It is noteworthy that the microrobot's instantaneous velocity accelerates in the middle of the path and decelerates at the ends, while the nanorobot exhibits a more square-wave-like pattern. The authors attribute this phenomenon to local intensity variations of the Gaussian beam spot. This is a very important observation that merits a deeper quantitative analysis. In steady-state motion, the robot's optical propulsion force is balanced by the viscous drag force. However, in a non-uniform Gaussian light field, the robot's equation of motion should account for the position-dependent propulsion force $F_{\text{thrust}}(I)$ and the optical gradient force $F_{\text{gradient}}(I,x)$. By establishing a more detailed physical model and fitting the experimental data to the theoretical model (which includes the Gaussian beam intensity profile), the physical parameters of the robot's interaction with the fluid environment could be extracted more accurately, thereby enhancing the persuasiveness of the conclusions.

We thank the reviewer for this insightful comment, which indeed provides an excellent opportunity to strengthen our analysis.

First, we discuss the optical gradient force acting on the robot in a Gaussian illumination profile. We recalculated this force using full-wave simulations by integrating Maxwell's stress tensor around the robot, as shown in Fig. R8. In Fig. R8a, we present the simulation geometry: the microrobot is propelled in the +y direction by the photon-recoil thrust F_o generated by its plasmonic motors, while the optical gradient force F_{grad} , arising from the non-uniform Gaussian beam, points toward the beam center. Figure R8b shows the position-dependent gradient force obtained by laterally displacing either the microrobot or nanorobot along the y-axis. At an incident intensity of $0.3 \text{ mW}/\mu\text{m}^2$, the gradient force is below 4 fN—two orders of magnitude smaller than the thrust force ($\sim 0.15 \text{ pN}$). This small magnitude is expected given the robots' subwavelength dimensions and refractive indices close to that of water.

Figure R8. Simulations for Optical gradient force. a Simulation geometry. b Position-dependent optical gradient force for micro- and nanorobots.

Next, we incorporate the position-dependent optical thrust resulting from the Gaussian intensity profile. The local thrust force can be written as:

$$F_o = \alpha I_o e^{-\frac{2r^2}{w^2}}$$

where α is the propulsion force per unit intensity (obtained from Fig. 3b), r is the radial distance from the beam center, and w is the beam waist. Because the optical gradient force is two orders of magnitude smaller, we omit it in the following analysis. The hydrodynamic drag force is modeled as:

$$F_{drag} = Dv$$

where D is the drag coefficient, which depends on both robot geometry and robot–substrate spacing. Using CFD simulations (COMSOL), we computed D for the nanorobot at different gap heights (Fig. R9b). For example, a nanorobot moving at 40 μm/s corresponds to a position ~4 μm from the beam center, allowing us to estimate the effective drag coefficient and the associated robot–substrate spacing. Figure R9c shows the predicted velocity as a function of position for a fixed drag coefficient. Consistent with experimental observations, the robot moves faster near the beam center—where the intensity is highest—and slower at the edges.

Figure R9 Motion dynamics in a non-uniform Gaussian beam. a Overlap between the Gaussian beam profile and measured rectangular trajectory. (The color in the trajectories indicates the instantaneous velocity.) b Predicted velocity–position dependence using the Gaussian thrust model. c Simulated drag coefficient for the nanorobot at various gap distances.

Furthermore, when the measured rectangular trajectory is overlaid onto the 2D Gaussian intensity map (Fig. R9a), the regions of high and low velocity align well with the intensity distribution: the nanorobot moves faster along the edges of the rectangle (closer to the beam center) and slows down significantly near the corners.

We will include this expanded discussion in revised manuscript (Page 6, Line 211-215) and the supporting figures in Supplementary Note S10.2.

Additionally, the paper demonstrates that the robot can maintain maneuverability while carrying a large bacterial load, which can weigh hundreds of times its own mass. This is an impressive phenomenon, but a corresponding quantitative analysis is lacking. The formation of a bacterial cluster significantly increases the robot's effective volume and mass, which in turn changes its viscous drag coefficient in the fluid. It is recommended that the authors provide quantitative data on the robot's motion performance under different bacterial loads in the Supplementary Information and discuss the relationship between the viscous drag coefficient and the load. This is crucial for evaluating its practical utility as a "nanoscale cleaner."

We thank the reviewer for this insightful comment. These points are addressed in detail in our response to **Reviewer 1, Comment #4**.

Briefly, when the robot carries a cluster of bacteria, its translational speed becomes limited by the increased hydrodynamic drag. As shown in Fig. R4b (Reviewer 1, #4), for a given target velocity there exists an upper limit to the number of bacteria that can be transported, although the robot can carry larger clusters at lower speeds. Importantly, the orientation control remains unaffected: even when loaded with a substantial bacterial cluster, the robot's orientation continues to follow the linear polarization reliably.

Language Optimization: In the abstract, the phrase "eliminates the need for beam steering and focusing, thereby reducing system complexity and photodamage" could be made more precise by replacing "reducing" with "mitigating," which better reflects the localized nature of the photothermal effect. Furthermore, the phrasing "The two-fold orientational locking degeneracy is lifted via spin-momentum transfer by circularly polarized light pulses" is somewhat awkward and could be simplified to "Circularly polarized light pulses are used to lift the two-fold orientational locking degeneracy via spin-momentum transfer." enters the correct stable alignment state each time.

We take reviewer's suggestion to make the changes in abstract.

Figure Consistency: The y-axis label "Reduced orientation (Degree)" in Fig. 2e requires a clearer definition to prevent ambiguity for the reader. The SEM images in Fig. 1a and Fig. 3a could adopt a more consistent style or labeling.

We have added more definition in the main text in **Page 5 (Line 172-175)**. For the SEM images in Fig. 1a, we chose to keep this presentation style to maintain clarity and clearly

display the entire robotic device. And in Fig. 2a and 3a, we provide zoomed-in views of the plasmonic motors and discuss the specific functionality of each component in detail.

Information Density: Some key experimental parameters (e.g., specific laser power, spot diameter, and EOM parameters used in different experiments) are scattered throughout the text, making it inconvenient for readers to find and replicate the experiments. It is suggested that the authors compile these key parameters into a single table and place it in the Methods or Supplementary Information section.

The spot size and EOM parameters are now summarized in **Supplementary Information S2**.

Deep Analysis and Clarification of the "Self-alignment" Mechanism

Comments	Contents in the manuscript
The paper proposes a mechanism that uses a restoring torque induced by linear polarized light to achieve orientational locking, which stably aligns the robot along the linear polarization axis to maximize propulsion efficiency. However, during turns, the authors state that a brief circularly polarized light pulse is required to achieve precise 90° turns. The need for this auxiliary pulse seems to contradict, to some extent, the core advantage of the "self-alignment" mechanism, which is to control direction using only linear polarized light. From a deeper physical perspective, this phenomenon reflects the dynamic relationship between the optical torque and the robot's intrinsic angular momentum and inertia. When the polarization state is switched rapidly (approx. 1 ms), the robot's moment of inertia may prevent it from immediately capturing the new stable alignment direction, instead causing it to settle in a metastable maximum of the orientational potential well. In this state, any minor disturbance, such as Brownian motion, could cause the robot to rotate randomly to the left or right, leading to a loss of deterministic control over the turning direction. The authors use a circularly polarized light pulse to provide an additional, deterministic torque via spin-momentum transfer, overcoming this uncertainty and ensuring the robot enters the correct stable alignment state each time.	The proposal of the "self-alignment" mechanism: Page 4, lines 112-113: Due to their elongated shape, the induced torque $T_z(\theta)$ is proportional to $\sin(2\theta)$, where θ is the angle between the nanorod's long axes and the linear polarization direction. Page 4, lines 117-119: Perpendicular alignment relative to the polarization corresponds to a metastable zero-torque state, which is highly sensitive to external disturbances such as Brownian motion. The 90° turn requires an explanation in terms of circularly polarized light: Page 3, Figure 1b's caption: At each corner, a short pulse of CW circularly polarized light is applied to enforce a right turn. Page 6, lines 151-155: However, because of the fast switching of the polarization at the corner and the disc's moment of inertia, the microrobot ends up in the metastable maximum of the OTP and a reorientation can occur in two ways leading to a 50% chance of a right turn instead of a left turn. Applying a circularly polarized pulse with the appropriate helicity provides additional torque to ensure a deterministic turn.

Although this turning mechanism is effective, its potential limitation lies in controlling more complex and smooth trajectories. The paper mentions in the discussion that “smooth paths can be implemented just by turning the linear polarization,” which implies two different control modes: one for straight motion and slow turns using pure linear polarization control, and another for precise 90° turns using a hybrid of linear and circular polarization. To make the paper more rigorous, the authors should add a detailed theoretical or simulation model in the Main text or Supplementary Information to quantify the dynamic response between rapid polarization switching and the robot's moment of inertia. Clearly differentiating the application scope of these two control modes and exploring how to achieve control of any complex trajectory using only a single polarization state (such as a rotatable linear polarization) by optimizing the motor design or control algorithm would greatly enhance the system's universality and appeal.

Discussion on the two control modes:

Page 6, Line 157: Smooth paths with only small rates of change of the propagation angle can be implemented just by turning the linear polarization.

Quantitative Analysis and Rigorous Scrutiny of the Bacterial Aggregation Mechanism

Comments	Contents in the manuscript
The paper primarily attributes the bacterial aggregation effect to thermophoresis and explicitly rules out the influence of optical trapping forces and convection. The authors estimate the bacterial Soret coefficient ST to be -2.45 K^{-1} by measuring the equilibrium diameter of bacterial clusters at different laser intensities. This conclusion provides valuable quantitative data for understanding the application of photothermal effects in biological manipulation. However, the paper does not provide sufficient experimental or simulation data to fully support its claim of "exclusivity."	The proposal of the thermal propulsion mechanism and the exclusion of other mechanisms: Page 10, lines 243-245: In contrast, the moderate local heating originating from plasmonic absorption causes the thermophoretic force F_t, which is primarily governed by the temperature gradient ∇T and provides a significantly extended trapping range. Page 11, lines 278-281: In a first step we excluded that optical trapping forces play a role in the assembly. We therefore carefully checked that no resonant optical fields, akin to whispering gallery modes [20], can form upon attachment of bacteria to the robot's body and that solely the thermophoretic force is responsible for the bacteria trapping and accumulation. Page 11, lines 300-302: Particle accumulation due to convective flow can be neglected as we confirm by finite element simulations. The predicted flow velocity is extremely small (below 2 nm/s), consistent with previous experimental findings. Estimation of the Soret coefficient: Page 11, Line 283: The thermophoretic force can be expressed as: $F_t = -k_B T S T \nabla T$, where ST is the Soret coefficient, k_B is the Boltzmann constant, and ∇T is the gradient of the temperature

	field. Page 11, lines 307-308: This happens when at the outermost rim of the cluster, the effective trapping potential due to the thermophoretic force is equal to $1k_B T$, equal to the average kinetic energy of bacteria undergoing Brownian motion. Page 12, lines 313-315: By identifying the trapping region's radius R_c corresponding to an energy threshold of $1k_B T$, we fit the simulation results to experimental data, yielding an estimated ST value of $-2.45K^{-1}$. Page 10, Figure 4e: Equilibrium bacterial cluster diameter as a function of applied laser intensity. Inset: Bacterial aggregation images at laser intensities of $0.06\text{mW}/\mu\text{m}^2$ (left top), $0.16\text{mW}/\mu\text{m}^2$ (left bottom), $0.28\text{mW}/\mu\text{m}^2$ (right top) and $0.4\text{mW}/\mu\text{m}^2$ (right bottom). f, Nanorobot transporting a single Escherichia coli bacterium along a '5'-shaped trajectory. Scale bar: $2\mu\text{m}$.
First, when multiple bacteria aggregate to form a micron-sized cluster, the cluster itself could act as a new optical scatterer or resonator, generating new optical forces in the near-field region. Although the optical gradient force on a single bacterium might be weak, the clustering effect could change the force distribution. Second, while the authors use finite-element simulations to rule out macroscopic convection, local fluid dynamics on the microscale (e.g., micro-convection caused by non-uniform temperature gradients) may still play a role in the initial capture and assembly process of bacteria. To enhance the persuasiveness of this conclusion, it is recommended that the authors:  1. Provide more detailed finite-element simulation results in the Supplementary Information, showing the light field distribution near the plasmonic motor and the corresponding optical 	

and thermophoretic force distributions in a multi-bacterial aggregation state. This would more intuitively demonstrate the dominant role of thermophoresis in long-range capture.

2. Conduct a more detailed analysis of the estimated Soret coefficient ST , comparing it with similar values from published literature and discussing the experimental uncertainties that might affect this value. For example, the non-spherical geometry of bacterial clusters and size differences between different bacterial types (e.g., *E. coli* and *S. carnosus*) could introduce estimation errors.

Comprehensive Analysis of Robot Motion Performance and Improvements in Data Visualization

Comments	Contents in the manuscript
The paper presents the movement of both the microrobot and nanorobot along rectangular trajectories and provides curves showing velocity as a function of laser intensity. It is noteworthy that the microrobot's instantaneous velocity accelerates in the middle of the path and decelerates at the ends, while the nanorobot exhibits a more square-wave-like pattern. The authors attribute this phenomenon to local intensity variations of the Gaussian beam spot. This is a very important observation that merits a deeper quantitative analysis. In steady-state motion, the robot's optical propulsion force is balanced by the viscous drag force. However, in a non-uniform Gaussian light field, the robot's equation of motion should account for the position-dependent propulsion force $F_{\text{thrust}}(I)$ and the optical gradient force $F_{\text{gradient}}(I,x)$. By establishing a more detailed physical model and fitting the experimental data to the theoretical model (which includes the Gaussian beam intensity profile), the physical parameters of the robot's interaction with the fluid environment could be extracted more accurately, thereby enhancing the persuasiveness of the conclusions.	Description of the speed variation curve: Micron-sized robot: Page 5, Line 145: it accelerates until reaching the middle of the path and then decelerates, an effect caused by local laser intensity variations.  Nanorobot: Page 8, Lines 208-209: The instantaneous velocity exhibits a square-wave-like pattern, where each peak represents translational motion along the side of the rectangle (inset in Fig. 3d). 
	The initial explanation for the cause of the speed variation: Page 8, lines 199-202: This is because, as the nanorobot moves through the aqueous solution, the optical thrust force is balanced by the viscous drag force. However, it is also subjected to the optical gradient force generated by the Gaussian laser beam, which has a more pronounced effect on larger objects. The paper mentioned the balance relationship among the thrust force, viscous resistance and the optical gradient force generated by the Gaussian beam, but no detailed motion equations were established for fitting.
Additionally, the paper demonstrates that the robot can maintain maneuverability while carrying a large bacterial load, which can weigh hundreds of times its own mass. This is an impressive phenomenon, but a corresponding quantitative analysis is lacking. The formation of a bacterial cluster significantly increases the robot's effective volume and mass, which in turn changes its viscous drag coefficient in the fluid. It is recommended that the authors provide quantitative data on the robot's motion performance under different bacterial loads in the Supplementary Information and discuss the relationship between the viscous drag coefficient and the load. This is crucial for evaluating its practical utility as a "nanoscale cleaner."	Description of the robot's load capacity: Page 11, lines 270-272: Despite carrying a bacterial load hundreds of times heavier than itself, the microrobot and its load remains fully maneuverable and the microrobot maintains its ability of orientational trapping along the direction of linear polarization. Page 12, lines 325-328: Interestingly, even with a bacterial load, the nanorobot maintains a remarkably high velocity, demonstrating the robustness of the driving mechanism in inhomogeneous environments and effective movement with a sufficiently low drag coefficient under different loading conditions. These are all qualitative descriptions and lack quantitative data support.

Paper Structure, Language, and Figure Presentation

Comments	Contents in the manuscript
Language Optimization: In the abstract, the phrase "eliminates the need for beam steering and focusing, thereby reducing system complexity and photodamage" could be made more precise by replacing "reducing" with "mitigating," which better reflects the localized nature of the photothermal effect. Furthermore, the phrasing "The two-fold orientational locking degeneracy is lifted via spin-momentum transfer by circularly polarized light pulses" is somewhat awkward and could be simplified to "Circularly polarized light pulses are used to lift the two-fold orientational locking degeneracy via spin-momentum transfer." enters the correct stable alignment state each time.	Abstract, line 13: thereby reducing system complexity and photodamage.” Abstract, Line 21: The two-fold orientational locking degeneracy is lifted via spin-momentum transfer by circularly polarized light pulses.
Figure Consistency: The y-axis label "Reduced orientation (Degree)" in Fig. 2e requires a clearer definition to prevent ambiguity for the reader. The SEM images in Fig. 1a and Fig. 3a could adopt a more consistent style or labeling.	Page 5, Figure 2e: The y-axis label is "Reduced orientation θ' (Degree)" Page 3, Figure 1a and Page 7, Figure 3a: These are the SEM images of micro robots and nanorobots respectively. A more consistent style or labeling can be adopted.
Information Density: Some key experimental parameters (e.g., specific laser power, spot diameter, and EOM parameters used in different experiments) are scattered throughout the text, making it inconvenient for readers to find and replicate the experiments. It is suggested that the authors compile these key parameters into a single table and place it in the Methods or Supplementary Information section.	The key parameters are scattered throughout the text, for example: Page 4, Line 137: Experimental parameters of t Page 8, lines 195-196: Experimental parameters of a Page 15, "Optical Setup and Measurement" section: Describes the optical setup, but there is no centralized parameter table.